# CROPPABLE KNOWLEDGE GRAPH EMBEDDING

## ABSTRACT

Knowledge Graph Embedding (KGE) is a common method for Knowledge Graphs (KGs) to serve various artificial intelligence tasks. The suitable dimensions of the embeddings depend on the storage and computing conditions of the specific application scenarios. Once a new dimension is required, a new KGE model needs to be trained from scratch, which greatly increases the training cost and limits the efficiency and flexibility of KGE in serving various scenarios. In this work, we propose a novel KGE training framework MED, through which we could train once to get a croppable KGE model applicable to multiple scenarios with different dimensional requirements, sub-models of the required dimensions can be cropped out of it and used directly without any additional training. In MED, we propose a mutual learning mechanism to improve the low-dimensional sub-models performance and make the high-dimensional sub-models retain the capacity that low-dimensional sub-models have, an evolutionary improvement mechanism to promote the high-dimensional sub-models to master the knowledge that the low-dimensional sub-models can not learn, and a dynamic loss weight to balance the multiple losses adaptively. Experiments on 4 KGE models over 4 standard KG completion datasets, 3 real application scenarios over a real-world large-scale KG, and the experiments of extending MED to the language model BERT show the effectiveness, high efficiency, and flexible extensibility of MED. The code and data are available at `https://anonymous.4open.science/r/MED-DBFC/`.

## 1 INTRODUCTION

Knowledge Graphs (KGs) are composed of triples representing facts in the form of (*head entity, relation, tail entity*), abbreviated as (*h, r, t*). KG has been widely used in recommendation systems Zhu et al. (2021); Zhang et al. (2021), information extraction Hoffmann et al. (2011); Daiber et al. (2013), question answering Zhang et al. (2016); Diefenbach et al. (2018) and other tasks. A common way to apply a knowledge graph is to represent the entities and relations in the knowledge graph into continuous vector spaces, called knowledge graph embedding (KGE) Bordes et al. (2013); Sun et al. (2019b), and then use the vector representation of entities and relations to serve a variety of tasks.

KGEs with higher dimensions have greater expressive power and usually achieve better performance, but this also means a larger number of parameters and requires more storage space and computing resources Zhu et al. (2022); Sachan (2020). The appropriate dimensions of the KGE are different for different devices or scenarios. As shown in Fig. 1, large remote servers have large storage space and sufficient computing resources to support high-dimensional KGE with good performance, while small and medium-sized terminal devices, such as vehicle-mounted systems or smartphones, can only accept low-dimensional KGE due to limited computing power and storage capacity. Therefore, according to the conditions of different devices or scenes, people tend to train the KGE with appropriate dimensions and as high quality as possible. However, the challenge is that once a new dimension is required, a new KGE needs to be trained from scratch. Especially when only low-dimensional KGE can be applied, to ensure good performance, the additional model compression technology such as knowledge distillation Hinton et al. (2015); Zhu et al. (2022) is needed during training. This significantly increases training costs and limits KGE's efficiency and flexibility in serving different scenarios.

Thus a new concept "croppable KGE" is proposed and we are interested in the research question that **is it possible to train a croppable KGE, with which KGEs of various required dimensions can**

**be cropped out of it, directly be used without any additional training, and achieve promising performance?**

Figure 1: Diverse KGE dimensions for a KG.

In this work, our main idea of croppable KGE learning is to train an entire KGE that contains many sub-models of different dimensions in it. These sub-models share their embedding parameters and are trained simultaneously. The goal is that the low-dimensional sub-models can benefit from the more expressive high-dimensional sub-models, while the high-dimensional sub-models retain the ability of the low-dimensional sub-models and master the knowledge that the low-dimensional sub-models cannot. Based on this idea, we propose a croppable KGE train-ing framework **MED**, which consists of three main modules, the **M**utual learning mechanism, the **E**volutionary improvement mechanism, and the **D**ynamic loss weight to achieve the above purpose. Specifically, the **mutual learning mechanism** is based on knowledge distillation and it makes pair-wise neighbor sub-models learn from each other, so that the performance of the lower-dimensional sub-model can be improved, and the higher-dimensional sub-model can retain the ability of the lower-dimensional sub-model. The **evolutionary improvement mechanism** helps the high-dimensional sub-model master more knowledge that the low-dimensional sub-model cannot by making the high-dimensional sub-model pay more attention to learn the triples that the low-dimensional sub-model can't correctly predict. The **dynamic loss weight** is designed to adaptively balance multiple losses of different sub-models according to their dimensions and further improve the overall performance.

We evaluate the effectiveness of our proposed MED by implementing it on three typical KGE methods and four standard KG datasets. We also prove its practical value by applying MED to a real-world large-scale KG and downstream tasks. Furthermore, we demonstrate the extensibility of MED by implementing it on language model BERT Devlin et al. (2019) and GLUE Wang et al. (2019) benchmarks. The experimental results show that (1) MED successfully trains a croppable KGE model available for various dimensional requirements, which contains multiple parameter-shared sub-models of different dimensions that of high performance and can be used directly without additional training; (2) the training efficiency of MED is far higher than that of independently training multiple KGE models of different sizes or obtaining them by knowledge distillation. (3) MED can be flexibly extended to other neural network models besides KGE and achieve good performance; (4) our proposed mutual learning mechanism, evolutionary improvement mechanism, and dynamic loss weight are effective and necessary for MED to achieve overall optimal performance. In summary, our contributions are as follows:

- We propose a new research question and task: training croppable KGE, from which KGEs of different dimensions can be cropped and used directly without any additional training.

- We propose a novel framework MED, including a mutual learning mechanism, an evolution-ary improvement mechanism, and a dynamic loss weight, to ensure the overall performance of all sub-models during training the croppable KGE.

- We experimentally prove that all sub-models of MED work well, especially the performance of the low-dimensional sub-models exceeding the KGE with the same dimension trained by the state-of-the-art distillation-based methods. MED also shows excellent performance in real-world applications and good extensibility on other types of neural networks.

## 2 RELATED WORK

This work is to achieve a croppable KGE that meets different dimensional requirements. One of the most common methods to obtain a good-performance KGE of the target dimension is utilizing knowledge distillation with a high-dimensional powerful teacher KGE. Thus, we focus on two research fields most relevant to our work: knowledge graph embedding and knowledge distillation.

## 2.1 Knowledge Graph Embedding

Knowledge graph embedding (KGE) technology has been widely applied with the key idea of mapping entities and relations of a KG into continuous vector spaces as vector representations, which can further serve various KG downstream tasks. TransE Bordes et al. (2013) is the most representative translation-based KGE method by regarding the relation as a translation from the head to tail entity. Variants of TransE include TransH Wang et al. (2014), TransR Lin et al. (2015), TransD Ji et al. (2015) and so on. RESCAL Nickel et al. (2011) is the first one based on vector decomposition, and then to improve it, DistMult Yang et al. (2015), ComplEx Trouillon et al. (2016), and SimplE Kazemi & Poole (2018) are proposed. RotatE Sun et al. (2019b) is a typical rotation-based method that regards the relation as the rotation between the head and tail entities. QuatE Zhang et al. (2019) and DihEdral Xu & Li (2019) work with a similar idea. PairRE Chao et al. (2021) uses two relation vectors to project the head and tail entities into an Euclidean space to encode complex relational patterns. With the development of neural networks, KGEs based on graph neural networks (GNNs) Dettmers et al. (2018); Nguyen et al. (2018); Schlichtkrull et al. (2018); Vashishth et al. (2020) are also proposed. Although the KGEs are simple and effective, there is an obvious challenge: In different scenarios, the required KGE dimensions are different, which depends on the storage and computing resources of the device. It has to train a new KGE model from scratch for a new dimension requirement, which greatly increases the training cost and limits the flexibility for KGE to serve diversified scenarios.

## 2.2 Knowledge Distillation

High-dimensional KGEs have strong expression ability due to the large number of parameters, but require a lot of storage and computing resources, and are not suitable for all scenarios, especially small devices. To solve this problem, a common way is to compress a high-dimensional KGE to the target low-dimensional KGE by knowledge distillation Hinton et al. (2015); Mirzadeh et al. (2020) and quantization Bai et al. (2021); Stock et al. (2021) technology.

Quantization replaces continuous vector representations with lower-dimensional discrete codes. TS-CL Sachan (2020) is the first work of KGE compression applying quantization. LightKG Wang et al. (2021a) uses a residual module to induce diversity among codebooks. However, quantization cannot improve the inference speed so it's still not suitable for devices with limited computing resources.

Knowledge distillation (KD) has been widely used in Computer Vision Mirzadeh et al. (2020) and Natural Language Processing Devlin et al. (2019); Sun et al. (2019a), helping reduce the model size and increase the inference speed. The core idea is to use the output of a large teacher model to guide the training of a small student model. DualDE Zhu et al. (2022) is a representative KD-based work to transfer the knowledge of high-dimensional KGE to low-dimensional KGE. It considers the mutual influences between the teacher and student and finetunes the teacher during training. MulDE Wang et al. (2021b) transfers the knowledge from multiple low-dimensional teacher models to a student model for hyperbolic KGE. ISD Zhou et al. (2022b) improves low-dimensional KGE by making it play the teacher and student roles alternatively during training. IterDE Liu et al. (2023) introduces an iterative distillation way and enables a KGE model to be the student and teacher during distilling alternately, thus knowledge can be transferred smoothly between high-dimensional teacher and low-dimensional student. Other distillation works related to knowledge graph include PMD Fan et al. (2024) applying distillation to pre-trained language models to improve KG completion, IncDE Liu et al. (2024) using distillation between the same-dimensional models at different times for incremental learning, and SKDE Xu et al. (2024) proposing self-knowledge distillation to avoid introducing a complex teacher model. Among these methods, DualDE Zhu et al. (2022) and IterDE Liu et al. (2023) are more relevant to our work, all have the setting that compresses high-dimensional teacher into low-dimensional student model. In this work, we propose a novel KD-based KGE training framework MED, one training can obtain a croppable KGE that meets multiple dimensional requirements.

## 3 Preliminary

Knowledge graph embedding (KGE) methods aim to express the relations between entities in a continuous vector space through a scoring function $f$. Specifically, given a knowledge graph $\mathcal{G} = (\mathcal{E}, \mathcal{R}, \mathcal{T})$ where $\mathcal{E}$, $\mathcal{R}$ and $\mathcal{T}$ are the sets

Table 1: Score functions.

| KGE method | Scoring Function $f(\mathbf{h}, \mathbf{r}, \mathbf{t})$ |
|---|---|
| TransE Bordes et al. (2013) | $-\|\mathbf{h} + \mathbf{r} - \mathbf{t}\|$ |
| SimplE Kazemi & Poole (2018) | $\frac{1}{2}(<h^H, r, t^T> + <t^H, r^{-1}, h^T>)$ |
| RotatE Sun et al. (2019b) | $-\|\mathbf{h} \circ \mathbf{r} - \mathbf{t}\|$ |
| PairRE Chao et al. (2021) | $-\|\mathbf{h} \circ \mathbf{r}^H - \mathbf{t} \circ \mathbf{r}^T\|$ |

of entities, relations and all observed triples, we utilize the triple scoring function to measure the plausibility of triples in the embedding space for a triple $(h, r, t)$ where $h \in \mathcal{E}, r \in \mathcal{R}$ and $t \in \mathcal{E}$. The triple score function is denoted as $s_{(h,r,t)} = f(\mathbf{h}, \mathbf{r}, \mathbf{t})$ with embeddings of head entity $\mathbf{h}$, relation $\mathbf{r}$ and tail entity $\mathbf{t}$ as input. Table 1 summarizes the scoring functions of some popular KGE methods, where $\circ$ is the Hadamard product, $< x^1, ..., x^k > = \sum_i x_i^1 ... x_i^k$ is the generalized dot product. The higher the triple score, the more likely the model is to judge the triples as true. The optimization objective of KGE model is

$$L_{KGE} = - \sum_{(h,r,t) \in \mathcal{T} \cup \mathcal{T}^-} y \log \sigma(s_{(h,r,t)}) + (1 - y) \log(1 - \sigma(s_{(h,r,t)})), \quad (1)$$

where $\mathcal{T}^- = \mathcal{E} \times \mathcal{R} \times \mathcal{E} \setminus \mathcal{T}$ is the set of negative triples, $\sigma$ is the Sigmoid activation function, $y$ is the ground-truth label of triple $(h, r, t)$, $y = 1$ for positive triples and $y = 0$ for negative triples.

## 4 MED FRAMEWORK

As shown in Fig. 2, our croppable KGE framework MED contains multiple (let's say $n$) sub-models of different dimensions in it, denoted as $M_i(i = 1, 2..., n)$ with dimension of $d_i$. Each sub-model $M_i$ is composed of the first $d_i$ dimensions of the whole embedding and the score of triple $(h, r, t)$ output by $M_i$ is $s_{(h,r,t)}^i = f(\mathbf{h}[0{:}d_i], \mathbf{r}[0{:}d_i], \mathbf{t}[0{:}d_i])$, where $\mathbf{h}[0{:}d_i]$ represents the first $d_i$ elements of vector $\mathbf{h}$. The parameters of sub-model $M_i$ are shared by all sub-models $M_j(i{<}j{\leqslant}n)$ that are higher-dimensional than it. The number of sub-models $n$ and the specific dimension of each sub-model $d_i$ can be set according to the actual application needs. For low-dimensional sub-models, we want to improve their performance as much as possible. For high-dimensional sub-models, we hope they cover the abilities that low-dimensional sub-models already have and master the knowledge that low-dimensional sub-models can not learn well, that is, they need to correctly predict not only the

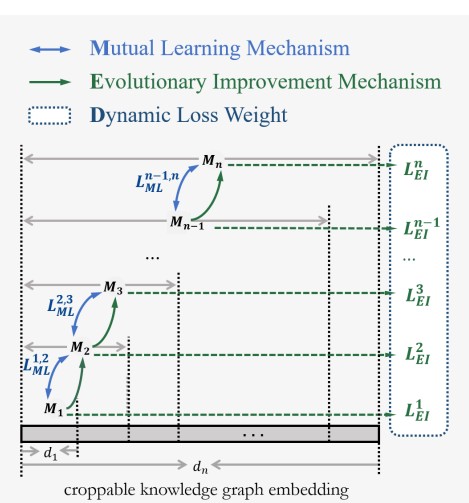

Figure 2: Overview of MED.

triples that low-dimensional sub-models can predict correctly but also those low-dimensional sub-models predict wrongly.

MED is based on knowledge distillation Hinton et al. (2015); Tang et al. (2019); Devlin et al. (2019) technique that the student learns by fitting the hard (ground-truth) label and the soft label from the teacher simultaneously. In MED, we first propose a *mutual learning mechanism* that makes low-dimensional sub-models learn from high-dimensional sub-models to achieve better performance, and makes high-dimensional sub-models also learn from low-dimensional sub-models to retain the abilities that low-dimensional sub-models already have. Then, we propose an *evolutionary improvement mechanism* to enable high-dimensional sub-models to master the knowledge that the low-dimensional sub-models can not learn well. Finally, we train MED with *dynamic loss weight* to adaptively balance multiple optimization objectives of sub-models.

### 4.1 MUTUAL LEARNING MECHANISM

We treat each sub-model $M_i$ as the student of its higher-dimensional neighbor sub-model $M_{i+1}$ to achieve better performance, since high-dimensional KGEs usually have more expressive power than low-dimensional ones due to more parameters Sachan (2020); Zhu et al. (2022). We also treat sub-model $M_i$ as the student of its lower-dimensional neighbor sub-model $M_{i-1}$, so the higher-dimensional sub-model can review what the lower-dimensional sub-model has learned and retain the low-dimensional one's existing abilities. Thus, pairwise neighbor sub-models serve as both teachers and students, learning from each other. The mutual learning loss between each pair of neighbor

sub-models is

$$L_{ML}^{i-1,i} = \sum_{(h,r,t)\in\mathcal{T}\cup\mathcal{T}^-} d_\delta\left(s_{(h,r,t)}^{i-1}, s_{(h,r,t)}^i\right), 1 < i \leqslant n, \quad (2)$$

where $s_{(h,r,t)}^i$ is the score of triple $(h,r,t)$ output by sub-model $M_i$ and reflects the possibility that this triplet exists, $\mathcal{T}^- = \mathcal{E}\times\mathcal{R}\times\mathcal{E}\setminus\mathcal{T}$ is the negative triple set, $n$ is the number of sub-models, and $d_\delta$ is Huber loss Huber & Peter (1964) with $\delta = 1$ commonly used in knowledge distillation for KGE Zhu et al. (2022). MED makes each sub-model only learn from its neighbor sub-models. The advantage is that this not only reduces the computational complexity of training but also makes every pair of teacher and student models have a relatively small dimension gap, which is important and effective because the large gap of dimensions between teacher and student will destroy the distillation effect Mirzadeh et al. (2020); Zhu et al. (2022).

## 4.2 EVOLUTIONARY IMPROVEMENT MECHANISM

The hard (ground-truth) label is the other important supervision signal during training in knowledge distillation Hinton et al. (2015). High-dimensional sub-models need to master triples that low-dimensional sub-models can not learn well, that is, high-dimensional sub-models need to correctly predict those positive (negative) triples that are wrongly predicted to be negative (positive) by low-dimensional sub-models. In MED, for a given triple $(h,r,t)$, the optimization weight in sub-model $M_i$ for it depends on the triple score output by the previous sub-model $M_{i-1}$.

For a positive triple, the optimization weight of the model $M_i$ for it is negatively correlated with its score by the model $M_{i-1}$. Specifically, the higher its score from the model $M_{i-1}$ (meaning that $M_{i-1}$ has been able to correctly judge it as a positive sample), the lower the optimization weight of the model $M_i$ for it, and the lower its score from the model $M_{i-1}$ (meaning that $M_{i-1}$ wrongly judges it as a negative sample), the higher the optimization weight of the model $M_i$ for it because $M_{i-1}$ cannot predict this triple well. The optimization weight of $M_i$ for the positive triple is

$$pos_{h,r,t}^i = \frac{\exp w_1/s_{(h,r,t)}^{i-1}}{\sum_{(h,r,t)\in T_{batch}}\exp w_1/s_{(h,r,t)}^{i-1}} \text{ if } 1 < i \leqslant n\,;\quad \frac{1}{|T_{batch}|} \text{ if } i = 1, \quad (3)$$

where $s_{(h,r,t)}^{i-1}$ is the score for triple $(h,r,t)$ output by the sub-model $M_{i-1}$, $T_{batch}$ is the set of positive triples within a batch, and $w_1$ is a learnable scaling parameter. Conversely, for a negative triple, the optimization weight of the model $M_i$ for it is positively correlated with its score by the model $M_{i-1}$ Sun et al. (2019b), and the optimization weight of $M_i$ for the negative triple is

$$neg_{h,r,t}^i = \frac{\exp w_2\cdot s_{(h,r,t)}^{i-1}}{\sum_{(h,r,t)\in T_{batch}^-}\exp w_2\cdot s_{(h,r,t)}^{i-1}} \text{ if } 1 < i \leqslant n\,;\quad \frac{1}{|T_{batch}^-|} \text{ if } i = 1, \quad (4)$$

where $T_{batch}^-$ is the set of negative triples within a batch, and $w_2$ is a learnable scaling parameter.

Therefore, the evolutionary improvement loss of the sub-model $M_i$ is

$$L_{EI}^i = -\sum_{(h,r,t)\in\mathcal{T}\cup\mathcal{T}^-} pos_{h,r,t}^i\cdot y\log\sigma(s_{(h,r,t)}^i) + neg_{h,r,t}^i\cdot(1-y)\log(1-\sigma(s_{(h,r,t)}^i)), \quad (5)$$

where $\sigma$ is the Sigmoid activation function, $y$ is the ground-truth label of the triple $(h,r,t)$, and it is 1 for positive triples and 0 for negative ones. In each sub-model, different hard (ground-truth) label loss weights are set for different triples, and the high-dimensional sub-model will pay more attention to learn the triple that the low-dimensional sub-model can not learn well.

## 4.3 DYNAMIC LOSS WEIGHT

Since MED involves the optimization of multiple sub-models, we set dynamic loss weights during training. Initially, low-dimensional sub-models prioritize learning from high-dimensional sub-models to improve performance. This means low-dimensional sub-models rely more on soft label information, so for low-dimensional sub-models, evolutionary improvement loss should account for less than mutual learning loss. Conversely, high-dimensional sub-models should focus more on capturing knowledge that low-dimensional models lack, while mitigating the impact of low-quality outputs from

low-dimensional models to maintain their good performance, that is, high-dimensional sub-models rely more on hard label information. So for high-dimensional sub-models, evolutionary improvement loss should account for more than mutual learning loss. For a teacher-student pair, their mutual learning loss acts on both teacher and student models simultaneously, so the effect of mutual learning loss for them is theoretically the same. We set different evolutionary improvement loss weights for different sub-models, and the final training loss function of MED is

$$L = \sum_{i=2}^{n} L_{ML}^{i-1,i} + \sum_{i=1}^{n} \exp(\frac{w_3 \cdot d_i}{d_n}) \cdot L_{EI}^{i}, \tag{6}$$

where $w_3$ is a learnable scaling parameter, and $d_i$ is the dimension of the $i$th sub-model.

## 5 EXPERIMENT

We evaluate MED on typical KGE and GLUE benchmarks and particularly answer the following research questions: (**RQ1**) Is it capable for MED to train a croppable KGE at once that multiple sub-models of different dimensions can be cropped from it and all achieve promising performance? (**RQ2**) Can MED finally achieve parameter-efficient KGE models? (**RQ3**) Does MED work in real-world applications? (**RQ4**) Can MED be extended to other neural networks besides KGE?

### 5.1 EXPERIMENT SETTING

#### 5.1.1 DATASET AND KGE METHODS

MED is universal and can be applied to any KGE method with a triple score function, we select three commonly used KGE methods as examples: TransE Bordes et al. (2013), SimplE Kazemi & Poole (2018), RotatE Sun et al. (2019b) and PairRE Chao et al. (2021), the triple score functions are described in Table 1.

We conduct comparison experiments on two common KG completion benchmark datasets WN18RR Toutanova et al. (2015) and FB15K237 Dettmers et al. (2018) and two more larger-scale KGs CoDEx-L Safavi & Koutra (2020) and YAGO3-10 Mahdisoltani et al. (2015). Besides, we apply our MED on a real-world large-scale e-commerce social knowledge

Table 2: Statistics of datasets.

| Dataset | #Ent. | #Rel. | #Train | #Valid | #Test |
|---------|-------|-------|--------|--------|-------|
| WN18RR | 40,943 | 11 | 86,835 | 3,034 | 3,134 |
| FB15K237 | 14,541 | 237 | 272,115 | 17,535 | 20,466 |
| CoDEx-L | 77,951 | 69 | 551,193 | 30,622 | 30,622 |
| YAGO3-10 | 123,143 | 37 | 1,079,040 | 4,978 | 4,982 |
| SKG | 6,974,959 | 15 | 50,775,620 | - | - |

graph (SKG) involving more than 50 million triples of social records by about 7 million users in the Taobao platform in real application scenarios. Table 2 shows the statistics of the datasets.

#### 5.1.2 EVALUATION METRIC

For the link prediction task, we adopt standard metrics MRR and Hit@$k$ ($k = 1, 3, 10$) in the filtered setting Bordes et al. (2013). For a test triple $(h, r, t)$, we construct candidate triples by replacing $h$ with all entities and keeping the replaced triples not in training, validation, and test set. Then we calculate the triple score rank of $(h, r, t)$ among all candidate triples as its head prediction rank $rank_t$. Similarly, we get its tail prediction rank $rank_t$. We average $rank_h$ and $rank_t$ as $(h, r, t)$'s final rank. MRR is the mean reciprocal rank of all test triples, and Hit@$k$ is the percentage of test triples with rank $\leq k$. We use *Effi* Chen et al. (2023), that is *MRR/#P* (*#P* is the number of parameters), to quantify the parameter efficiency of models. We use f1-score and accuracy for user labeling task, and normalized discounted cumulative gain ndcg@$k$($k = 5, 10$) for product recommendation task.

#### 5.1.3 IMPLEMENTATION

For the link prediction task, we set $d_n = 640$ for the highest-dimensional sub-model $M_n$ and $d_1 = 10$ for the lowest-dimensional sub-model $M_1$. We set $n = 64$ and the dimension gap 10 for every pair of neighbor sub-models. There are a total of 64 available sub-models of different dimensions from 10 to 640 in our croppable KGE model. The dimension of sub-model $M_i(i = 1, 2..., 64)$ is $10 \times i$. For the user labeling and product recommendation task, we set $n = 3$ and train the croppable KGE containing 3 sub-models: $M_1$ with $d_1 = 10$ for mobile phone (MB) terminals that are limited by storage and

computing resources, $M_2$ with $d_2 = 100$ for the personal computer (PC), and $M_3$ with $d_3 = 500$ for the platform's servers. We initialize the learnable scaling parameters $w_i, w_2$ and $w_3$ in equation 3, equation 4 and equation 6 to 1. We implement MED by extending OpenKE Han et al. (2018), an open-source KGE framework based on PyTorch. We set the batch size to $1024$ and the maximum training epoch to 3000 with early stopping. For each positive triple, we generate 64 negative triples by randomly replacing its head or tail entity with another entity. We use Adam Kingma & Ba (2015) optimizer with a linear decay learning rate scheduler and perform a search on the initial learning rate in $\{0.0001, 0.0005, 0.001, 0.01\}$. We train all sub-models simultaneously by optimizing the uniformly sampled sub-models from the full Croppable model in each step.

### 5.1.4 BASELINES

For each required dimension $d_r$, we extract the first $d_r$ dimensions from our croppable KGE as the target model and compare it to the KGE models obtained by 8 baselines of the following 3 types:

- Directly training the target KGE model of requirement dimension $d_r$, referred to as **1) DT**. The directly trained highest-dimensional KGE model ($d_r = d_n$) is marked as $M_{max}^{DT}$.

- Extracting the first $d_r$ dimensions from $M_{max}^{DT}$ as the target model, referred to as **2) Ext**. Besides, we update $M_{max}^{DT}$ by assessing the importance of each one of 640 dimensions and arranging them in descending order before extracting as Molchanov et al. (2017); Voita et al. (2019): **3) Ext-L**, the importance for each dimension of $M_{max}^{DT}$ is the variation of KGE loss on validation set after removing it; and **4) Ext-V**, the importance for each dimension is the average absolute of its parameter weights of all entities and all relations.

- Distilling the target KGE by KD methods: **5) BKD** Hinton et al. (2015) is the most basic one by minimizing the KL divergence of the output distributions of teacher and student; **6) TA** Mirzadeh et al. (2020) uses a medium-size teaching assistant (TA) model as a bridge for size gap, where TA model has the same dimension as the directly trained one whose MRR is closest to the average MRR of teacher and student. We also compare with two KD methods proposed for KGE, which have similar configurations to ours, i.e. compressing high-dimensional teacher into low-dimensional student: **7) DualDE** Zhu et al. (2022) considers the mutual influences between teacher and student and optimizes them simultaneously; **8) IterDE** Liu et al. (2023) enables the KGE model to alternately act as student and teacher so that knowledge can be transferred smoothly between high-dimensional teacher and low-dimensional student. In these baselines, $M_{max}^{DT}$ is the teacher, and other settings including hyperparameters are the same as their original papers.

### 5.2 PERFORMANCE COMPARISON

We report the link prediction results of some representative dimensions in Table 3, more results of other dimensions and metrics are in Appendix A and the ablation studies are in Appendix B.

MED outperforms baselines in almost all settings, especially for the extremely low dimensions. On WN18RR with $d$=10, MED achieves an improvement of **14.9%** and **15.1%** on TransE, **8.4%** and **6.6%** on RotatE, **29.4%** and **10.6%** on PairRE compared with the best MRR and Hit@10 of baselines. We can observe a similar phenomenon on FB15K237. This benefits from the rich knowledge sources of low-dimensional models in MED: For sub-model $M_i$, $M_{i+1}$ is the teacher directly next to it, while $M_{i+2}$ can also indirectly affect $M_i$ by directly affecting $M_{i+1}$. Theoretically, all higher-dimensional sub-models can finally transfer their knowledge to low-dimensional sub-models through stepwise propagation. Although such stepwise propagation may have negative effects on high-dimensional models by bringing low-quality knowledge from low-dimensional sub-models, the evolutionary improvement mechanism in MED weakens the damage and makes high-dimensional ones still achieve competitive performance than directly trained KGEs as in Fig. 3. We also find that Ext-based methods perform extremely unstable: Ext, Ext-L, and Ext-V work worse than DT except on WN18RR with TransE, indicating that only considering the importance of each dimension is not enough to guarantee the performance of all sub-models. More results and ablation studies are in Appendix A and Appendix B.

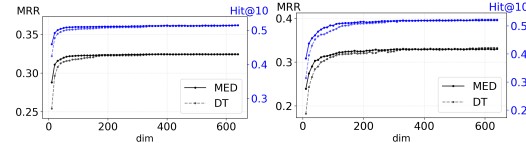

Figure 3: Results of different dimensions for PairRE on WN18RR (left) and FB15K237 (right).

Table 3: MRR and Hit@10 (H10) of some dimensions on WN18RR (WN) and FB15K237 (FB).

| | | WN18RR | | | | | | | | FB15K237 | | | | | | | |
|---|---|---|---|---|---|---|---|---|---|---|---|---|---|---|---|---|---|
| | | 10d | | 40d | | 160d | | 640d | | 10d | | 40d | | 160d | | 640d | |
| *KGE* | *Method* | *MRR* | *H10* | *MRR* | *H10* | *MRR* | *H10* | *MRR* | *H10* | *MRR* | *H10* | *MRR* | *H10* | *MRR* | *H10* | *MRR* | *H10* |
| | DT | 0.121 | 0.287 | 0.214 | 0.496 | 0.233 | 0.531 | 0.237 | 0.537 | 0.150 | 0.235 | 0.299 | 0.477 | 0.315 | 0.499 | 0.322 | **0.508** |
| | Ext | 0.125 | 0.298 | 0.199 | 0.468 | 0.225 | 0.515 | 0.237 | 0.537 | 0.115 | 0.211 | 0.236 | 0.392 | 0.286 | 0.462 | 0.322 | 0.508 |
| | Ext-L | 0.139 | 0.315 | 0.224 | 0.497 | 0.236 | **0.534** | 0.237 | 0.537 | 0.109 | 0.194 | 0.232 | 0.381 | 0.285 | 0.462 | 0.322 | 0.508 |
| | Ext-V | 0.139 | 0.309 | 0.222 | 0.494 | 0.236 | 0.532 | 0.237 | 0.537 | 0.139 | 0.256 | 0.237 | 0.396 | 0.293 | 0.466 | 0.322 | 0.508 |
| TransE | BKD | 0.141 | 0.323 | 0.226 | 0.513 | 0.233 | 0.531 | - | - | 0.176 | 0.293 | 0.303 | 0.480 | 0.315 | 0.501 | - | - |
| | TA | 0.144 | 0.335 | 0.226 | 0.512 | 0.234 | 0.533 | - | - | 0.175 | 0.246 | 0.303 | 0.484 | 0.319 | 0.504 | - | - |
| | DualDE | 0.148 | 0.337 | 0.225 | 0.514 | 0.235 | 0.533 | - | - | 0.179 | 0.301 | 0.306 | 0.483 | 0.319 | 0.505 | - | - |
| | IterDE | 0.143 | 0.332 | 0.224 | 0.511 | 0.236 | 0.531 | - | - | 0.176 | 0.285 | 0.307 | 0.482 | 0.317 | 0.505 | - | - |
| | MED | **0.170** | **0.388** | **0.232** | **0.518** | **0.236** | 0.529 | **0.237** | **0.537** | **0.196** | **0.341** | **0.308** | **0.486** | **0.320** | 0.505 | **0.322** | 0.507 |
| | DT | 0.061 | 0.126 | 0.316 | 0.389 | 0.409 | 0.459 | 0.421 | 0.481 | 0.097 | 0.179 | 0.236 | 0.390 | 0.285 | 0.458 | **0.295** | **0.472** |
| | Ext | 0.004 | 0.007 | 0.160 | 0.249 | 0.357 | 0.401 | 0.421 | 0.481 | 0.037 | 0.068 | 0.090 | 0.144 | 0.229 | 0.372 | 0.295 | 0.472 |
| | Ext-L | 0.005 | 0.006 | 0.169 | 0.244 | 0.398 | 0.454 | 0.421 | 0.481 | 0.045 | 0.059 | 0.083 | 0.146 | 0.196 | 0.316 | 0.295 | 0.472 |
| | Ext-V | 0.004 | 0.006 | 0.246 | 0.317 | 0.398 | 0.461 | 0.421 | 0.481 | 0.049 | 0.069 | 0.105 | 0.149 | 0.224 | 0.369 | 0.295 | 0.472 |
| SimplE | BKD | 0.075 | 0.156 | 0.343 | 0.399 | 0.414 | 0.468 | - | - | 0.113 | 0.204 | 0.244 | 0.412 | 0.287 | 0.463 | - | - |
| | TA | 0.089 | 0.189 | 0.368 | 0.418 | 0.415 | 0.472 | - | - | 0.124 | 0.221 | 0.254 | 0.416 | 0.290 | 0.465 | - | - |
| | DualDE | 0.083 | 0.175 | **0.386** | 0.423 | **0.419** | 0.475 | - | - | 0.120 | 0.213 | 0.258 | **0.429** | 0.293 | 0.466 | - | - |
| | IterDE | 0.077 | 0.162 | 0.375 | 0.419 | 0.416 | 0.469 | - | - | 0.120 | 0.215 | 0.257 | 0.427 | **0.293** | 0.465 | - | - |
| | MED | **0.111** | **0.224** | 0.385 | **0.431** | 0.418 | **0.477** | **0.421** | **0.482** | **0.143** | **0.267** | **0.261** | 0.427 | 0.291 | **0.466** | 0.294 | 0.470 |
| | DT | 0.172 | 0.418 | 0.456 | 0.556 | 0.471 | 0.567 | 0.476 | 0.575 | 0.254 | 0.424 | 0.312 | 0.495 | 0.322 | 0.506 | **0.325** | **0.515** |
| | Ext | 0.299 | 0.378 | 0.437 | 0.516 | 0.467 | 0.549 | 0.476 | **0.575** | 0.138 | 0.245 | 0.251 | 0.410 | 0.291 | 0.465 | 0.325 | 0.515 |
| | Ext-L | 0.206 | 0.277 | 0.399 | 0.487 | 0.445 | 0.541 | 0.476 | 0.575 | 0.135 | 0.243 | 0.221 | 0.365 | 0.280 | 0.453 | 0.325 | 0.515 |
| | Ext-V | 0.261 | 0.377 | 0.337 | 0.471 | 0.416 | 0.532 | 0.476 | 0.575 | 0.160 | 0.281 | 0.238 | 0.393 | 0.288 | 0.458 | 0.325 | 0.515 |
| RotatE | BKD | 0.175 | 0.434 | 0.457 | 0.556 | 0.472 | 0.570 | - | - | 0.277 | 0.442 | 0.314 | 0.503 | 0.322 | 0.510 | - | - |
| | TA | 0.177 | 0.438 | 0.459 | 0.558 | 0.473 | 0.572 | - | - | 0.280 | 0.447 | 0.313 | 0.501 | 0.323 | 0.510 | - | - |
| | DualDE | 0.179 | 0.440 | 0.462 | 0.559 | **0.473** | 0.573 | - | - | 0.282 | 0.449 | 0.315 | 0.502 | 0.322 | **0.512** | - | - |
| | IterDE | 0.176 | 0.436 | 0.459 | 0.560 | 0.471 | 0.569 | - | - | 0.276 | 0.445 | 0.317 | 0.504 | 0.323 | 0.512 | - | - |
| | MED | **0.324** | **0.469** | **0.466** | **0.561** | 0.471 | **0.574** | 0.476 | 0.574 | **0.288** | **0.459** | **0.318** | **0.504** | 0.323 | 0.510 | 0.324 | 0.514 |
| | DT | 0.220 | 0.321 | 0.415 | 0.472 | 0.449 | 0.534 | **0.453** | **0.544** | 0.182 | 0.314 | 0.284 | 0.452 | 0.319 | 0.505 | **0.332** | **0.522** |
| | Ext | 0.152 | 0.209 | 0.334 | 0.463 | 0.419 | 0.526 | 0.453 | 0.544 | 0.148 | 0.222 | 0.217 | 0.353 | 0.294 | 0.469 | 0.332 | 0.522 |
| | Ext-L | 0.162 | 0.220 | 0.363 | 0.442 | 0.437 | 0.523 | 0.453 | 0.544 | 0.150 | 0.249 | 0.219 | 0.333 | 0.309 | 0.489 | 0.332 | 0.522 |
| | Ext-V | 0.172 | 0.260 | 0.389 | 0.456 | 0.441 | 0.529 | 0.453 | 0.544 | 0.176 | 0.277 | 0.229 | 0.374 | 0.311 | 0.490 | 0.332 | 0.522 |
| PairRE | BKD | 0.228 | 0.336 | 0.421 | 0.483 | 0.451 | 0.536 | - | - | 0.198 | 0.332 | 0.288 | 0.453 | 0.321 | 0.508 | - | - |
| | TA | 0.245 | 0.340 | 0.426 | 0.487 | 0.452 | 0.537 | - | - | 0.208 | 0.346 | 0.292 | 0.455 | 0.323 | 0.509 | - | - |
| | DualDE | 0.242 | 0.336 | 0.428 | 0.495 | **0.453** | 0.540 | - | - | 0.207 | 0.342 | 0.293 | 0.456 | **0.326** | **0.512** | - | - |
| | IterDE | 0.235 | 0.336 | 0.426 | 0.495 | 0.450 | 0.538 | - | - | 0.205 | 0.340 | 0.293 | 0.462 | 0.324 | 0.508 | - | - |
| | MED | **0.317** | **0.376** | **0.433** | **0.502** | 0.451 | **0.541** | 0.451 | 0.542 | **0.239** | **0.384** | **0.303** | **0.466** | 0.324 | 0.510 | 0.330 | 0.520 |

## 5.3 PARAMETER EFFICIENCY OF MED

In Table 4, we compare our sub-models of suitable low dimensions to parameter-efficient KGEs especially proposed for large-scale KGs including NodePiece Galkin et al. (2022) and EARL Chen et al. (2023). In the case that the number of model parameters is roughly equivalent, the performance of the sub-models of MED exceeds that of the specialized parameter-efficient KGE methods. This demonstrates sub-models of our method are parameter efficient. More importantly, it can provide parameter-efficient models of different size for applications.

Table 4: Link prediction results on WN18RR, FB15K237, CoDEx-L and YAGO3-10.

| | FB15k-237 | | | | | WN18RR | | | | | CoDEx-L | | | | | YAGO3-10 | | | | |
|---|---|---|---|---|---|---|---|---|---|---|---|---|---|---|---|---|---|---|---|---|
| | *Dim* | *#P(M)* | *MRR* | *Hit@10* | *Effi* | *Dim* | *#P(M)* | *MRR* | *Hit@10* | *Effi* | *Dim* | *#P(M)* | *MRR* | *Hit@10* | *Effi* | *Dim* | *#P(M)* | *MRR* | *Hit@10* | *Effi* |
| RotatE | 1000 | 29.3 | 0.336 | 0.532 | 0.011 | 500 | 40.6 | 0.508 | 0.612 | 0.013 | 500 | 78 | 0.258 | 0.387 | 0.003 | 500 | 123.2 | 0.495 | 0.670 | 0.004 |
| RotatE | 100 | 2.9 | 0.296 | 0.473 | 0.102 | 50 | 4.1 | 0.411 | 0.429 | 0.100 | 25 | 3.8 | 0.196 | 0.322 | 0.052 | 20 | 4.8 | 0.121 | 0.262 | 0.025 |
| + NodePiece | 100 | 3.2 | 0.256 | 0.420 | 0.080 | 100 | 4.4 | 0.403 | 0.515 | 0.092 | 100 | 3.6 | 0.190 | 0.313 | 0.053 | 100 | 4.1 | 0.247 | 0.488 | 0.060 |
| + EARL | 150 | 1.8 | 0.310 | 0.501 | 0.172 | 200 | 3.8 | 0.440 | 0.527 | 0.116 | 100 | 2.1 | 0.238 | **0.390** | 0.113 | 100 | 3 | 0.302 | 0.498 | **0.101** |
| + MED | 40 | 1.2 | 0.318 | 0.504 | 0.265 | 40 | 3.2 | **0.466** | **0.561** | 0.146 | 20 | 3.1 | **0.243** | 0.385 | 0.078 | 20 | 4.9 | **0.313** | **0.528** | 0.064 |

## 5.4 MED IN REAL APPLICATIONS

We apply the trained croppable KGE with TransE on SKG to three real applications: the user labeling task on servers and the product recommendation task on PCs and mobile phones. Table 5 shows that our croppable user embeddings substantially exceed all baselines includ-

Table 5: Results on SKG.

| | | User Labeling | | Product Recommendation | | | |
|---|---|---|---|---|---|---|---|
| | | server (500d) | | PC terminal (100d) | | MP terminal (10d) | |
| *Method* | *train time* | *acc.* | *f1* | *ndcg@5* | *ndcg@10* | *ndcg@5* | *ndcg@10* |
| DT | 103h | 0.889 | 0.874 | 0.411 | 0.441 | 0.344 | 0.361 |
| PCA | - | - | - | 0.417 | 0.447 | 0.392 | 0.418 |
| DualDE | 195h | - | - | 0.423 | 0.456 | 0.404 | 0.433 |
| MED | 53h | **0.893** | **0.879** | **0.431** | **0.465** | **0.422** | **0.451** |

ing directly trained (DT), the best baseline DualDE, and a common dimension reduction method in industry principal components analysis (PCA) on $M_{max}^{DT}$. Notably, the excellent performance on the mobile phone task (which can only carry embeddings with a maximum dimension of 10 limited by storage and computing resources) demonstrates the enormous practical value of our approach. More application details are in Appendix C.

## 5.5 EXTEND MED TO NEURAL NETWORKS

To verify the extensibility of our method to other neural networks, we take the language model BERT Devlin et al. (2019) as an example. We uniformly adopt distillation methods implemented

based on Hugging Face Transformers Wolf et al. (2020) as baselines. Following previous works Sun et al. (2019a); Tang et al. (2019); Jung et al. (2023); Zhou et al. (2022a), we distill at the fine-tuning stage. More experimental details are in Appendix D.

Table 6: Results on the dev set of GLUE. The results of knowledge distillation methods for $BERT_4$ and $BERT_6$ are reported by Jung et al. (2023); Zhou et al. (2022a) and the [†]results reported by us.

| Method | #P(M) | Speedup | MNLI-m acc. | MNLI-mm acc. | MRPC f1/acc. | QNLI acc. | QQP f1/acc. | RTE acc. | STS-2 acc. | STS-B pear./spear. |
|---|---|---|---|---|---|---|---|---|---|---|
| $BERT^{†}_{Base}$ | 110 | 1.0× | 84.4 | 85.3 | 88.6/84.1 | 89.7 | 89.6/91.1 | 67.5 | 92.5 | 88.8/88.5 |
| $BERT_6$-BKD | 66 | 2.0× | 82.2 | 82.9 | 86.2/80.8 | 88.5 | 88.0/91.0 | 65.4 | 90.9 | 82.2/87.8 |
| $BERT_6$-PKD | 66 | 2.0× | 82.3 | 82.6 | 86.4/81.0 | 88.6 | 87.9/91.0 | 63.9 | 90.8 | 88.5/88.1 |
| $BERT_6$-MiniLM | 66 | 2.0× | 82.2 | 82.6 | 84.6/78.1 | **89.5** | 87.2/90.5 | 61.5 | 90.2 | 87.8/87.5 |
| $BERT_6$-RKD | 66 | 2.0× | 82.4 | 82.9 | 86.9/81.8 | 88.9 | 88.1/**91.2** | 65.2 | 91.0 | 88.4/88.1 |
| $BERT_6$-FSD | 66 | 2.0× | 82.4 | 83.0 | 87.1/82.2 | 89.0 | 88.1/**91.2** | 66.6 | 91.0 | **88.7/88.3** |
| $BERT_4$-BKD | 55 | 2.9× | 80.5 | 80.9 | 87.2/83.1 | 87.5 | 86.6/90.4 | 65.2 | 90.2 | 84.5/84.2 |
| $BERT_4$-PKD | 55 | 2.9× | 80.9 | 81.3 | 87.0/82.9 | 87.7 | 86.8/90.5 | 66.1 | 90.5 | 84.3/84.0 |
| $BERT_4$-MetaDistil | 55 | 2.9× | 82.4 | 82.7 | **88.4/84.2** | 88.6 | 87.8/90.8 | **67.8** | 91.8 | 86.3/86.0 |
| BERT-HAT[†] | 54 | 2.0× | 70.8 | 71.6 | 81.2/74.8 | 65.3 | 76.1/80.4 | 52.7 | 84.3 | 79.6/80.1 |
| BERT-MED | 54 | 2.0× | **82.7** | **83.3** | 88.0/84.0 | 86.8 | **89.1**/90.7 | 67.2 | **91.9** | 87.6/87.2 |
| BERT-HAT[†] | 17.5 | 4.7× | 63.6 | 64.2 | 68.4/78.4 | 61.1 | 69.0/79.7 | 47.2 | 82.9 | 74.1/75.8 |
| BERT-MED | 17.5 | 4.7× | 81.2 | 82.4 | 86.1/82.0 | 86.4 | 83.8/86.2 | 64.6 | 88.2 | 86.1/86.4 |
| BERT-HAT[†] | 6.36 | 5.2× | 59.9 | 60.0 | 66.5/77.3 | 60.1 | 66.5/77.1 | 46.2 | 81.7 | 71.9/70.4 |
| BERT-MED | 6.36 | 5.2× | 72.6 | 73.7 | 84.1/78.1 | 86.0 | 79.6/82.7 | 61.7 | 86.9 | 82.8/81.6 |

Table 6 shows the results on the development set of GLUE Wang et al. (2019). We compare MED with other KD models under similar speedup or a comparable number of parameters. MED achieves competitive performance on most tasks compared to BERT-specialized KD methods. In addition, when compared to HAT Wang et al. (2020a), which shares the most similar model architecture to ours, sub-models of MED outperform HAT across three different parameter quantities. Specifically, sub-models with 54M, 17.5M, and 6.36M parameters achieve average 16.3%, 21.7% and 19.7% improvements respectively.

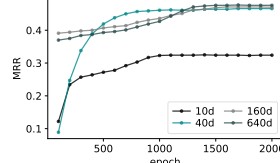

Figure 4: Sub-models' MRR during training on WN18RR with RotatE.

### 5.6 ANALYSIS OF MED

#### 5.6.1 TRAINING EFFICIENCY

We report the training time of obtaining 64 models of all sizes ($d$=10, 20, ..., 640) by different methods in Table 7. Figure 4 showing how the MRR of different sub-models changes during training. For DT, the training time cost is the sum of the time of directly training 64 KGE models of all sizes in turn. For the Ext-based baselines, the training time cost is the same and is equal to the time of training a $d_n$-

Table 7: Training time (hours).

| | | TransE | | SimplE | | RotatE | | PairRE | |
|---|---|---|---|---|---|---|---|---|---|
| WN | DT | 74.0 | (9.49×) | 68.0 | (12.14×) | 141.0 | (11.10×) | 67.4 | (10.06×) |
| | Ext-based | 1.5 | (0.19×) | 1.3 | (0.23×) | 2.5 | (0.20×) | 1.6 | (0.24×) |
| | BKD | 91.5 | (11.73×) | 72.0 | (12.86×) | 163.0 | (12.83×) | 87.5 | (13.06×) |
| | TA | 172.0 | (22.05×) | 142.0 | (25.36×) | 272.0 | (21.42×) | 166.0 | (24.78×) |
| | DualDE | 151.0 | (19.36×) | 133.0 | (23.75×) | 240.0 | (18.90×) | 133.0 | (19.85×) |
| | IterDE | 140.9 | (18.06×) | 118.0 | (21.07×) | 216.0 | (17.01×) | 124.0 | (18.51×) |
| | MED | 7.8 | (1.00×) | 5.6 | (1.00×) | 12.7 | (1.00×) | 6.7 | (1.00×) |
| FB | DT | 218.0 | (10.23×) | 179.0 | (10.65×) | 381.0 | (10.73×) | 179.0 | (9.37×) |
| | Ext-based | 4.7 | (0.22×) | 5.1 | (0.30×) | 9.5 | (0.27×) | 3.7 | (0.19×) |
| | BKD | 248.0 | (11.64×) | 227.0 | (13.51×) | 443.0 | (12.48×) | 231.0 | (12.09×) |
| | MED | 21.3 | (1.00×) | 16.8 | (1.00×) | 35.5 | (1.00×) | 19.1 | (1.00×) |

dimensional KGE model since the time of arranging dimensions is very short and negligible. For the KD-based baselines, the training time cost is the sum of the time of training the $d_n$-dimensional teacher model and distilling 63 student models ($d$=10, 20, ..., 630) in turn. All training is performed on a single NVIDIA Tesla A100 40GB GPU for fair comparison. For TA, DualDE and IterDE on FB15K237, we don't train student models of all 63 sizes, which is estimated to take more than 400 hours on each KGE method. Compared with directly trained (DT) models of all sizes in turn, MED accelerates by up to 10× for 4 KGE methods. Although Ext-based baselines spend the shortest training time, they perform particularly poorly and lack practical value. Except for BKD, KD-based methods need to optimize both the student model and larger teacher model, which greatly increases the training parameters and time cost.

#### 5.6.2 EFFECT OF THE NUMBER OF SUB-MODELS

We set the number of different sub-models, i.e. $n$= 64, 16, 4 on WN18RR with RotatE. And Table 8 shows that when the number of sub-models is reduced, the performance of high-dimensional ($d$=160 and 640)

Table 8: Results of different $n$.

| $n$ | train time | 10d MRR | H10 | 40d MRR | H10 | 160d MRR | H10 | 640d MRR | H10 |
|---|---|---|---|---|---|---|---|---|---|
| 64 | 12.7h | 0.324 | 0.469 | 0.466 | 0.561 | 0.471 | 0.574 | 0.476 | 0.574 |
| 16 | 6.2h | 0.322 | 0.467 | 0.465 | 0.561 | 0.473 | 0.575 | 0.477 | 0.576 |
| 4 | 3.3h | 0.319 | 0.463 | 0.463 | 0.561 | 0.475 | 0.577 | 0.480 | 0.578 |

models improves, while the performance of low-dimensional ($d$=10 and 40) models decreases (still exceeds the best result of baselines in Table 3 that MRR=0.299 of Ext with $d$=10, MRR=0.462 of DualDE with $d$=40). The training efficiency is almost linearly related to the number of models.

### 5.6.3 WHETHER HIGH-DIMENSIONAL SUB-MODELS COVER THE CAPABILITIES OF LOW-DIMENSIONAL ONES

If a high-dimensional model retains the ability of lower-dimensional models, it should correctly predict all triples that the lower-dimensional model can predict. We count the percentage of triples in test set that meet the condition that if the smallest sub-model that can correctly predict a given triple is $M_i$, all higher-dimensional sub-models ($M_{i+1}$, $M_{i+2}$, ..., $M_n$) also correctly predict it, and denote the result as the ability retention ratio (ARR). We use Hit@10 to judge whether a triple is correctly predicted, that is, $M_i$ correctly predicts a triple if $M_i$ scores this triple in the top 10 among all candidate triples.

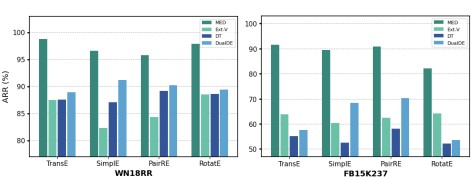

Figure 5: The ability retention ratio (ARR).

From Fig. 5, ARR of MED is always much higher than baselines, especially on FB15K237, indicating that high-dimensional sub-models in MED successfully cover the power of low-dimensional ones, contributed by the mutual learning mechanism that helps high-dimensional sub-models review what low-dimensional sub-models have learned. Based on this advantage of MED, we can provide a simple way to judge how easy or difficult a triple is for KGE methods to learn: the triple that low-dimensional sub-models can correctly predict may be easy since high-dimensional models can also predict it, while triples that can only be predicted by high-dimensional sub-models are difficult.

### 5.6.4 VISUAL ANALYSIS OF EMBEDDING

We select four primary entity categories ('organization', 'sports', 'location', and 'music') that contain more than 300 entities in FB15K237, and randomly select 250 entities for each. We cluster these entities' embeddings of 3 different dimensions ($d$=10, 100, 600) by the t-SNE algorithm, and the clustering results are visualized in Fig. 6. Under the same dimension, the clustering result of MED is always the best, followed by DualDE, while the result of Ext-V is generally poor, which is consistent with the conclusion in

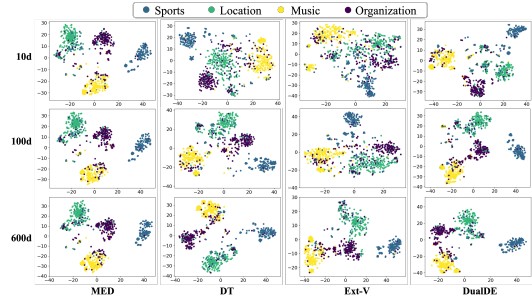

Figure 6: Clustering on FB15K237 with RotatE.

Section 5.2. We also find some special phenomenons for MED when dimension increases: 1) the nodes of the 'sports' gradually become two clusters meaning MED learns more fine-grained category information as dimension increases. and 2) the relative distribution among different categories hardly changes and shows a trend of "inheritance" and "improvement". This further proves MED achieves our expectation that high-dimensional sub-models retain the ability of low-dimensional sub-models, and can learn more knowledge than low-dimensional sub-models.

## 6 CONCLUSION

In this work, we propose a novel KGE training framework, MED, that trains a croppable KGE at once, and then sub-models of various required dimensions can be cropped out from it and used directly without additional training. In MED, we propose the mutual learning mechanism to improve low-dimensional sub-models performance and make the high-dimensional sub-models retain the ability of the low-dimensional ones, the evolutionary improvement mechanism to motivate high-dimensional sub-models to master more knowledge that low-dimensional ones cannot, and the dynamic loss weight to adaptively balance multiple losses. The experimental results show the effectiveness and high efficiency of our method, where all sub-models achieve promising performance, especially the performance of low-dimensional sub-models is greatly improved. In future work, we will further explore the more fine-grained information encoding ability of each sub-model.

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

## A   MORE RESULTS OF LINK PREDICTION

More results of link prediction are shown in Table 9 and Table 10 for WN18RR, and Table 11 and Table 12 for FB15K237. All comparison results of sub-models of MED to the directly trained KGEs (DT) of 10- to 640-dimension are shown in Fig. 7.

Table 9: MRR and Hit@10 of some representative dimensions on WN18RR.

| | Method | 10d MRR | 10d Hit@10 | 20d MRR | 20d Hit@10 | 40d MRR | 40d Hit@10 | 80d MRR | 80d Hit@10 | 160d MRR | 160d Hit@10 | 320d MRR | 320d Hit@10 | 640d MRR | 640d Hit@10 |
|---|---|---|---|---|---|---|---|---|---|---|---|---|---|---|---|
| **TransE** | DT | 0.121 | 0.287 | 0.176 | 0.453 | 0.214 | 0.496 | 0.227 | 0.524 | 0.233 | 0.531 | 0.235 | 0.534 | **0.237** | **0.537** |
| | Ext | 0.125 | 0.298 | 0.172 | 0.423 | 0.199 | 0.468 | 0.213 | 0.495 | 0.225 | 0.515 | 0.226 | 0.521 | 0.237 | 0.537 |
| | Ext-L | 0.139 | 0.315 | 0.196 | 0.461 | 0.224 | 0.497 | 0.232 | 0.516 | 0.236 | **0.534** | 0.236 | 0.535 | 0.237 | 0.537 |
| | Ext-V | 0.139 | 0.309 | 0.198 | 0.458 | 0.222 | 0.494 | 0.234 | 0.525 | 0.236 | 0.532 | 0.236 | 0.536 | 0.237 | 0.537 |
| | BKD | 0.141 | 0.323 | 0.207 | 0.480 | 0.226 | 0.513 | 0.232 | 0.527 | 0.233 | 0.531 | 0.236 | 0.533 | - | - |
| | TA | 0.144 | 0.335 | 0.211 | 0.483 | 0.226 | 0.512 | 0.233 | 0.527 | 0.234 | 0.533 | 0.236 | 0.535 | - | - |
| | DualDE | 0.148 | 0.337 | 0.213 | 0.488 | 0.225 | 0.514 | **0.234** | **0.530** | 0.235 | 0.533 | **0.238** | 0.535 | - | - |
| | IterDE | 0.143 | 0.332 | 0.211 | 0.484 | 0.224 | 0.511 | 0.232 | 0.528 | 0.236 | 0.531 | 0.237 | 0.533 | - | - |
| | MED | **0.170** | **0.388** | **0.219** | **0.491** | **0.232** | **0.518** | 0.232 | 0.523 | **0.236** | 0.529 | 0.237 | **0.536** | 0.237 | 0.537 |
| **SimplE** | DT | 0.061 | 0.126 | 0.257 | 0.372 | 0.316 | 0.389 | 0.382 | 0.446 | 0.409 | 0.459 | 0.417 | 0.474 | **0.421** | 0.481 |
| | Ext | 0.004 | 0.007 | 0.051 | 0.107 | 0.160 | 0.249 | 0.219 | 0.314 | 0.357 | 0.401 | 0.407 | 0.451 | 0.421 | 0.481 |
| | Ext-L | 0.005 | 0.006 | 0.048 | 0.078 | 0.169 | 0.244 | 0.369 | 0.435 | 0.398 | 0.454 | 0.417 | 0.481 | 0.421 | 0.481 |
| | Ext-V | 0.004 | 0.006 | 0.047 | 0.076 | 0.246 | 0.317 | 0.368 | 0.402 | 0.398 | 0.461 | 0.413 | 0.472 | 0.421 | 0.481 |
| | BKD | 0.075 | 0.156 | 0.285 | 0.381 | 0.343 | 0.399 | 0.394 | 0.450 | 0.414 | 0.468 | 0.418 | 0.475 | - | - |
| | TA | 0.089 | 0.189 | 0.316 | 0.386 | 0.368 | 0.418 | 0.405 | 0.456 | 0.415 | 0.472 | 0.421 | 0.481 | - | - |
| | DualDE | 0.083 | 0.175 | 0.328 | 0.388 | **0.386** | **0.423** | 0.407 | 0.454 | **0.419** | 0.475 | **0.422** | **0.482** | - | - |
| | IterDE | 0.077 | 0.162 | 0.321 | 0.378 | 0.375 | 0.419 | 0.404 | 0.452 | 0.416 | 0.469 | 0.421 | 0.482 | - | - |
| | MED | **0.111** | **0.224** | **0.335** | **0.395** | 0.385 | **0.431** | **0.407** | **0.457** | 0.418 | **0.477** | 0.421 | 0.481 | 0.421 | **0.482** |
| **RotatE** | DT | 0.172 | 0.418 | 0.409 | 0.504 | 0.456 | 0.556 | 0.465 | 0.564 | 0.471 | 0.567 | 0.474 | 0.573 | **0.476** | **0.575** |
| | Ext | 0.299 | 0.378 | 0.379 | 0.464 | 0.437 | 0.516 | 0.458 | 0.544 | 0.467 | 0.549 | 0.471 | 0.552 | 0.476 | 0.575 |
| | Ext-L | 0.206 | 0.277 | 0.336 | 0.424 | 0.399 | 0.487 | 0.423 | 0.515 | 0.445 | **0.541** | 0.466 | 0.564 | 0.476 | 0.575 |
| | Ext-V | 0.261 | 0.377 | 0.304 | 0.433 | 0.337 | 0.471 | 0.366 | 0.497 | 0.416 | 0.532 | 0.451 | 0.561 | 0.476 | 0.575 |
| | BKD | 0.175 | 0.434 | 0.424 | 0.540 | 0.457 | 0.556 | 0.471 | 0.565 | 0.472 | 0.570 | 0.474 | 0.572 | - | - |
| | TA | 0.177 | 0.438 | 0.424 | 0.542 | 0.459 | 0.558 | 0.470 | 0.567 | 0.473 | 0.572 | 0.474 | 0.572 | - | - |
| | DualDE | 0.179 | 0.440 | 0.425 | 0.542 | 0.462 | 0.559 | **0.471** | 0.567 | **0.473** | 0.573 | 0.475 | 0.573 | - | - |
| | IterDE | 0.176 | 0.436 | 0.421 | 0.538 | 0.459 | 0.560 | 0.470 | 0.567 | 0.471 | 0.569 | 0.474 | 0.572 | - | - |
| | MED | **0.324** | **0.469** | **0.456** | **0.543** | **0.466** | **0.561** | 0.471 | **0.568** | 0.471 | **0.574** | **0.476** | 0.573 | 0.476 | 0.574 |
| **PairRE** | DT | 0.220 | 0.321 | 0.342 | 0.381 | 0.415 | 0.472 | 0.435 | 0.516 | 0.449 | 0.534 | 0.452 | 0.542 | **0.453** | **0.544** |
| | Ext | 0.152 | 0.209 | 0.261 | 0.379 | 0.334 | 0.463 | 0.375 | 0.493 | 0.419 | 0.526 | 0.438 | 0.545 | 0.453 | 0.544 |
| | Ext-L | 0.162 | 0.220 | 0.281 | 0.360 | 0.363 | 0.442 | 0.417 | 0.495 | 0.437 | 0.523 | 0.446 | 0.544 | 0.453 | 0.544 |
| | Ext-V | 0.172 | 0.260 | 0.306 | 0.374 | 0.389 | 0.456 | 0.420 | 0.498 | 0.441 | 0.529 | 0.446 | 0.541 | 0.453 | 0.544 |
| | BKD | 0.228 | 0.336 | 0.375 | 0.413 | 0.421 | 0.483 | 0.443 | 0.525 | 0.451 | 0.536 | 0.453 | 0.542 | - | - |
| | TA | 0.245 | 0.340 | 0.381 | 0.427 | 0.426 | 0.487 | 0.448 | 0.534 | 0.452 | 0.537 | 0.453 | 0.543 | - | - |
| | DualDE | 0.242 | 0.336 | 0.377 | 0.424 | 0.428 | 0.495 | **0.451** | 0.536 | **0.453** | 0.540 | **0.454** | **0.544** | - | - |
| | IterDE | 0.235 | 0.336 | 0.379 | 0.423 | 0.426 | 0.495 | 0.449 | 0.533 | 0.450 | 0.538 | 0.452 | 0.543 | - | - |
| | MED | **0.317** | **0.376** | **0.408** | **0.467** | **0.433** | **0.502** | 0.449 | **0.537** | 0.451 | **0.541** | 0.451 | 0.542 | 0.451 | 0.542 |

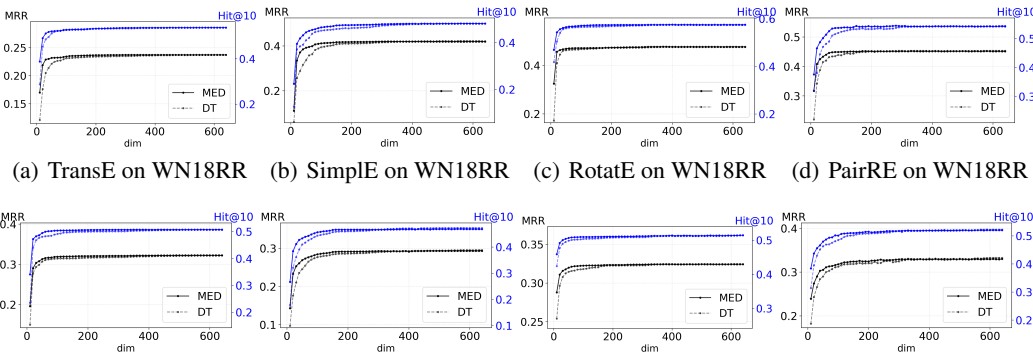

(a) TransE on WN18RR    (b) SimplE on WN18RR    (c) RotatE on WN18RR    (d) PairRE on WN18RR

(e) TransE on FB15K237    (f) SimplE on FB15K237    (g) RotatE on FB15K237    (h) PairRE on FB15K237

Figure 7: Performance of sub-models of MED and the directly trained (DT) KGEs of dimensions from 10 to 640.

# B   ABLATION STUDY

We conduct ablation studies to evaluate the effect of three modules in MED: the mutual learning mechanism (MLM), the evolutionary improvement mechanism (EIM), and the dynamic loss weight (DLW). Table 13 shows the MRR and Hit@$k$ ($k = 1, 3, 10$) of MED removing these modules respectively on WN18RR and TransE.

Table 10: Hit@3 and Hit@1 of some representative dimensions on WN18RR.

| | Method | 10d Hit@3 | 10d Hit@1 | 20d Hit@3 | 20d Hit@1 | 40d Hit@3 | 40d Hit@1 | 80d Hit@3 | 80d Hit@1 | 160d Hit@3 | 160d Hit@1 | 320d Hit@3 | 320d Hit@1 | 640d Hit@3 | 640d Hit@1 |
|---|---|---|---|---|---|---|---|---|---|---|---|---|---|---|---|
| **TransE** | DT | 0.202 | 0.011 | 0.291 | 0.016 | 0.385 | 0.018 | 0.401 | 0.025 | 0.403 | 0.027 | 0.407 | 0.033 | 0.412 | **0.034** |
| | Ext | 0.201 | 0.016 | 0.285 | 0.023 | 0.338 | 0.023 | 0.364 | 0.028 | 0.384 | 0.033 | 0.388 | 0.028 | 0.412 | 0.034 |
| | Ext-L | 0.218 | 0.029 | 0.317 | 0.025 | 0.361 | 0.039 | 0.403 | 0.046 | 0.405 | 0.036 | 0.408 | 0.033 | 0.412 | 0.034 |
| | Ext-V | 0.218 | 0.029 | 0.314 | **0.045** | 0.391 | **0.051** | 0.407 | **0.047** | 0.408 | 0.036 | 0.411 | 0.027 | 0.412 | 0.034 |
| | BKD | 0.216 | 0.035 | 0.331 | 0.040 | 0.392 | 0.033 | 0.401 | 0.031 | 0.404 | 0.030 | 0.407 | 0.032 | - | - |
| | TA | 0.224 | 0.040 | 0.343 | 0.043 | 0.395 | 0.037 | 0.408 | 0.030 | 0.407 | 0.030 | 0.410 | 0.034 | - | - |
| | DualDE | 0.226 | 0.037 | 0.346 | 0.043 | 0.394 | 0.037 | **0.408** | 0.031 | **0.408** | 0.031 | **0.411** | **0.034** | - | - |
| | IterDE | 0.217 | 0.032 | 0.345 | 0.044 | 0.392 | 0.036 | 0.407 | 0.030 | 0.408 | 0.031 | 0.407 | 0.033 | - | - |
| | MED | **0.269** | **0.040** | **0.369** | **0.045** | **0.399** | 0.038 | 0.404 | 0.042 | 0.407 | **0.037** | 0.410 | 0.033 | **0.412** | 0.031 |
| **SimplE** | DT | 0.061 | 0.028 | 0.297 | 0.193 | 0.361 | 0.289 | 0.406 | 0.343 | 0.420 | 0.382 | 0.428 | 0.386 | 0.433 | **0.391** |
| | Ext | 0.003 | 0.001 | 0.055 | 0.023 | 0.181 | 0.114 | 0.249 | 0.168 | 0.377 | 0.329 | 0.422 | 0.381 | 0.433 | 0.391 |
| | Ext-L | 0.004 | 0.003 | 0.051 | 0.031 | 0.187 | 0.128 | 0.389 | 0.333 | 0.413 | 0.365 | 0.429 | 0.384 | 0.433 | 0.391 |
| | Ext-V | 0.004 | 0.002 | 0.050 | 0.029 | 0.269 | 0.205 | 0.378 | 0.349 | 0.409 | 0.372 | 0.426 | 0.382 | 0.433 | 0.391 |
| | BKD | 0.077 | 0.034 | 0.331 | 0.225 | 0.384 | 0.311 | 0.415 | 0.358 | 0.426 | 0.371 | 0.431 | 0.385 | - | - |
| | TA | 0.093 | 0.042 | 0.349 | 0.269 | 0.375 | 0.349 | 0.412 | **0.384** | 0.425 | 0.388 | 0.431 | 0.389 | - | - |
| | DualDE | 0.086 | 0.038 | 0.361 | 0.285 | 0.391 | **0.368** | 0.416 | 0.383 | 0.427 | **0.389** | 0.434 | **0.392** | - | - |
| | IterDE | 0.079 | 0.033 | 0.355 | 0.279 | 0.382 | 0.356 | 0.415 | 0.379 | 0.424 | 0.383 | 0.433 | 0.389 | - | - |
| | MED | **0.119** | **0.048** | **0.366** | **0.292** | **0.395** | 0.359 | **0.419** | 0.380 | **0.429** | 0.389 | **0.435** | 0.391 | **0.434** | 0.390 |
| **RotatE** | DT | 0.304 | 0.005 | 0.436 | 0.357 | 0.475 | 0.393 | 0.487 | 0.420 | 0.489 | 0.423 | 0.491 | 0.428 | 0.493 | **0.429** |
| | Ext | 0.315 | 0.257 | 0.399 | 0.335 | 0.452 | 0.395 | 0.472 | 0.415 | 0.480 | 0.413 | 0.470 | 0.418 | 0.493 | 0.429 |
| | Ext-L | 0.224 | 0.166 | 0.359 | 0.288 | 0.420 | 0.352 | 0.441 | 0.373 | 0.461 | 0.396 | 0.481 | 0.417 | 0.493 | 0.429 |
| | Ext-V | 0.289 | 0.197 | 0.336 | 0.234 | 0.377 | 0.263 | 0.402 | 0.293 | 0.442 | 0.357 | 0.467 | 0.397 | 0.493 | 0.429 |
| | BKD | 0.312 | 0.009 | 0.452 | 0.361 | 0.479 | 0.403 | 0.487 | 0.421 | 0.490 | 0.424 | 0.492 | 0.425 | - | - |
| | TA | 0.314 | 0.010 | 0.452 | 0.363 | 0.481 | 0.408 | 0.489 | 0.420 | 0.488 | 0.422 | 0.492 | 0.425 | - | - |
| | DualDE | 0.320 | 0.011 | 0.452 | 0.364 | 0.483 | 0.412 | 0.489 | **0.423** | 0.488 | **0.426** | 0.491 | 0.425 | - | - |
| | IterDE | 0.311 | 0.013 | 0.439 | 0.356 | 0.479 | 0.407 | 0.484 | 0.423 | 0.488 | 0.425 | 0.493 | 0.424 | - | - |
| | MED | **0.354** | **0.277** | **0.476** | **0.409** | **0.486** | **0.418** | **0.490** | 0.422 | **0.492** | 0.424 | **0.493** | 0.427 | **0.495** | 0.428 |
| **PairRE** | DT | 0.271 | ,0.174 | 0.368 | 0.313 | 0.428 | 0.384 | 0.450 | 0.399 | 0.463 | 0.405 | 0.462 | 0.406 | **0.464** | **0.407** |
| | Ext | 0.163 | 0.120 | 0.292 | 0.198 | 0.366 | 0.267 | 0.398 | 0.314 | 0.437 | 0.364 | 0.452 | 0.388 | 0.464 | 0.407 |
| | Ext-L | 0.175 | 0.129 | 0.302 | 0.237 | 0.383 | 0.319 | 0.431 | 0.377 | 0.450 | 0.395 | 0.455 | 0.400 | 0.464 | 0.407 |
| | Ext-V | 0.192 | 0.124 | 0.323 | 0.269 | 0.407 | 0.352 | 0.435 | 0.379 | 0.452 | 0.398 | 0.458 | 0.400 | 0.464 | 0.407 |
| | BKD | 0.279 | 0.184 | 0.388 | 0.334 | 0.435 | 0.372 | 0.452 | 0.405 | 0.460 | 0.405 | 0.463 | 0.407 | - | - |
| | TA | 0.293 | 0.197 | 0.387 | 0.332 | 0.437 | 0.380 | 0.460 | 0.404 | 0.462 | 0.409 | 0.463 | 0.408 | - | - |
| | DualDE | 0.281 | 0.175 | 0.389 | 0.330 | 0.437 | 0.381 | **0.463** | **0.409** | 0.463 | 0.410 | 0.465 | **0.410** | - | - |
| | IterDE | 0.285 | 0.172 | 0.390 | 0.331 | 0.435 | 0.377 | 0.461 | 0.405 | 0.463 | **0.411** | 0.464 | 0.410 | - | - |
| | MED | **0.314** | **0.259** | **0.426** | **0.367** | **0.443** | **0.392** | 0.462 | 0.405 | **0.464** | 0.406 | **0.465** | 0.407 | 0.464 | 0.406 |

Table 11: MRR and Hit@10 of some representative dimensions on FB15K237.

| | Method | 10d MRR | 10d Hit@10 | 20d MRR | 20d Hit@10 | 40d MRR | 40d Hit@10 | 80d MRR | 80d Hit@10 | 160d MRR | 160d Hit@10 | 320d MRR | 320d Hit@10 | 640d MRR | 640d Hit@10 |
|---|---|---|---|---|---|---|---|---|---|---|---|---|---|---|---|
| **TransE** | DT | 0.150 | 0.235 | 0.277 | 0.440 | 0.299 | 0.477 | 0.313 | 0.484 | 0.315 | 0.499 | 0.318 | 0.501 | 0.322 | **0.508** |
| | Ext | 0.115 | 0.211 | 0.191 | 0.324 | 0.236 | 0.392 | 0.266 | 0.436 | 0.286 | 0.462 | 0.299 | 0.479 | 0.322 | 0.508 |
| | Ext-L | 0.109 | 0.194 | 0.175 | 0.293 | 0.232 | 0.381 | 0.263 | 0.424 | 0.285 | **0.462** | 0.301 | 0.484 | 0.322 | 0.508 |
| | Ext-V | 0.139 | 0.256 | 0.200 | 0.348 | 0.237 | 0.396 | 0.270 | 0.437 | 0.293 | 0.466 | 0.308 | 0.488 | 0.322 | 0.508 |
| | BKD | 0.176 | 0.293 | 0.279 | 0.446 | 0.303 | 0.480 | 0.315 | 0.500 | 0.315 | 0.501 | 0.320 | 0.502 | - | - |
| | TA | 0.175 | 0.246 | 0.281 | 0.441 | 0.303 | 0.484 | 0.314 | 0.498 | 0.319 | 0.504 | 0.321 | 0.504 | - | - |
| | DualDE | 0.179 | 0.301 | 0.281 | 0.443 | 0.306 | 0.483 | 0.316 | 0.502 | 0.319 | 0.505 | **0.322** | **0.508** | - | - |
| | IterDE | 0.176 | 0.285 | 0.276 | 0.446 | 0.307 | 0.482 | 0.315 | **0.503** | 0.315 | 0.505 | 0.319 | 0.505 | - | - |
| | MED | **0.196** | **0.341** | **0.290** | **0.472** | **0.308** | **0.486** | **0.317** | 0.502 | **0.320** | 0.505 | 0.321 | 0.507 | **0.322** | 0.507 |
| **SimplE** | DT | 0.097 | 0.179 | 0.176 | 0.321 | 0.236 | 0.390 | 0.271 | 0.431 | 0.285 | 0.458 | 0.291 | 0.467 | **0.295** | **0.472** |
| | Ext | 0.037 | 0.068 | 0.069 | 0.107 | 0.090 | 0.144 | 0.159 | 0.258 | 0.229 | 0.372 | 0.269 | 0.432 | 0.295 | 0.472 |
| | Ext-L | 0.045 | 0.059 | 0.056 | 0.062 | 0.083 | 0.146 | 0.114 | 0.205 | 0.196 | 0.316 | 0.258 | 0.421 | 0.295 | 0.472 |
| | Ext-V | 0.049 | 0.069 | 0.066 | 0.101 | 0.105 | 0.149 | 0.138 | 0.224 | 0.224 | 0.369 | 0.261 | 0.414 | 0.295 | 0.472 |
| | BKD | 0.113 | 0.204 | 0.182 | 0.315 | 0.244 | 0.412 | 0.275 | 0.439 | 0.287 | 0.463 | 0.293 | 0.470 | - | - |
| | TA | 0.124 | 0.221 | 0.192 | 0.329 | 0.254 | 0.416 | 0.276 | 0.448 | 0.290 | 0.465 | 0.295 | 0.471 | - | - |
| | DualDE | 0.120 | 0.213 | 0.195 | 0.346 | 0.258 | **0.429** | 0.279 | 0.443 | **0.293** | 0.466 | **0.296** | **0.468** | - | - |
| | IterDE | 0.120 | 0.215 | 0.193 | 0.338 | 0.257 | 0.427 | 0.281 | 0.440 | 0.293 | 0.465 | 0.297 | 0.468 | - | - |
| | MED | **0.143** | **0.267** | **0.233** | **0.384** | **0.261** | 0.427 | **0.279** | 0.448 | 0.291 | 0.466 | 0.293 | 0.468 | 0.294 | 0.470 |
| **RotatE** | DT | 0.254 | 0.424 | 0.297 | 0.477 | 0.312 | 0.495 | 0.317 | 0.502 | 0.322 | 0.506 | 0.323 | 0.510 | **0.325** | **0.515** |
| | Ext | 0.138 | 0.245 | 0.203 | 0.340 | 0.251 | 0.410 | 0.276 | 0.443 | 0.291 | 0.465 | 0.305 | 0.485 | 0.325 | 0.515 |
| | Ext-L | 0.135 | 0.243 | 0.188 | 0.319 | 0.221 | 0.365 | 0.246 | 0.402 | 0.280 | 0.453 | 0.299 | 0.477 | 0.325 | 0.515 |
| | Ext-V | 0.160 | 0.281 | 0.198 | 0.340 | 0.238 | 0.393 | 0.265 | 0.427 | 0.288 | 0.458 | 0.302 | 0.478 | 0.325 | 0.515 |
| | BKD | 0.277 | 0.442 | 0.305 | 0.485 | 0.314 | 0.503 | 0.321 | 0.508 | 0.322 | 0.510 | 0.323 | 0.509 | - | - |
| | TA | 0.280 | 0.447 | 0.306 | 0.485 | 0.313 | 0.501 | 0.319 | 0.507 | 0.323 | 0.510 | 0.323 | 0.509 | - | - |
| | DualDE | 0.282 | 0.449 | 0.307 | 0.486 | 0.315 | 0.502 | 0.318 | 0.507 | 0.322 | **0.512** | 0.324 | **0.514** | - | - |
| | IterDE | 0.276 | 0.445 | 0.306 | 0.482 | 0.317 | 0.504 | 0.319 | 0.508 | 0.323 | 0.512 | 0.324 | 0.513 | - | - |
| | MED | **0.288** | **0.459** | **0.311** | **0.492** | **0.318** | **0.504** | **0.322** | **0.509** | **0.323** | 0.510 | **0.324** | 0.512 | 0.324 | 0.514 |
| **PairRE** | DT | 0.182 | 0.314 | 0.243 | 0.395 | 0.284 | 0.452 | 0.307 | 0.476 | 0.319 | 0.505 | 0.328 | 0.518 | **0.332** | **0.522** |
| | Ext | 0.148 | 0.222 | 0.177 | 0.289 | 0.217 | 0.353 | 0.259 | 0.416 | 0.294 | 0.469 | 0.321 | 0.506 | 0.332 | 0.522 |
| | Ext-L | 0.150 | 0.249 | 0.196 | 0.294 | 0.219 | 0.333 | 0.271 | 0.436 | 0.309 | 0.489 | 0.326 | 0.513 | 0.332 | 0.522 |
| | Ext-V | 0.176 | 0.277 | 0.192 | 0.303 | 0.229 | 0.374 | 0.279 | 0.450 | 0.311 | 0.490 | 0.329 | 0.513 | 0.332 | 0.522 |
| | BKD | 0.198 | 0.332 | 0.251 | 0.407 | 0.288 | 0.453 | 0.311 | 0.487 | 0.321 | 0.508 | 0.330 | 0.521 | - | - |
| | TA | 0.208 | 0.346 | 0.263 | 0.430 | 0.292 | 0.455 | 0.314 | 0.493 | 0.323 | 0.509 | 0.332 | 0.521 | - | - |
| | DualDE | 0.207 | 0.342 | 0.261 | 0.427 | 0.293 | 0.456 | **0.316** | 0.495 | **0.326** | **0.512** | **0.334** | **0.524** | - | - |
| | IterDE | 0.205 | 0.340 | 0.264 | 0.431 | 0.293 | 0.462 | 0.314 | 0.494 | 0.324 | 0.508 | 0.332 | 0.522 | - | - |
| | MED | **0.239** | **0.384** | **0.274** | **0.437** | **0.303** | **0.466** | 0.314 | **0.495** | 0.324 | 0.510 | 0.329 | 0.521 | 0.330 | 0.520 |

## B.1 MUTUAL LEARNING MECHANISM (MLM)

We remove the mutual learning mechanism from MED and keep the other parts unchanged, where equation 6 is rewritten as

$$L = \sum_{i=1}^{n} \exp\left(\frac{w_3 \cdot d_i}{d_n}\right) \cdot L_{EI}^i. \tag{7}$$

From the result of "MED w/o MLM" in Table 13, we find that after removing the mutual learning mechanism, the performance of low-dimensional sub-models deteriorates seriously since the low-dimensional sub-models can not learn from the high-dimensional sub-models. For example, the MRR of the 10-dimensional sub-model decreased by $12.4\%$, and the MRR of the 20-dimensional sub-model decreased by $10\%$. While the performance degradation of the high-dimensional sub-model is not particularly obvious, and the MRR of the highest-dimensional sub-model ($dim = 640$) is not worse than that of MED, which is because to a certain degree, removing the mutual learning mechanism also avoids the negative influence to high-dimensional sub-models from low-dimensional sub-models. On the whole, this mechanism greatly improves the performance of low-dimensional sub-models.

## B.2 EVOLUTIONARY IMPROVEMENT MECHANISM (EIM)

In this part, we replace evolutionary improvement loss $L_{EI}^i$ in equation 6 with the regular KGE loss $L_{KGE}^i$:

$$L_{KGE}^i = \sum_{(h,r,t)\in\mathcal{T}\cup\mathcal{T}^-} y \log \sigma(s_{(h,r,t)}^i) + (1-y) \log(1 - \sigma(s_{(h,r,t)}^i)). \tag{8}$$

From the result of "MED w/o EIM" in Table 13, we find that removing the evolutionary improvement mechanism mainly degrades the performance of high-dimensional sub-models. While due to the

Table 12: Hit@3 and Hit@1 of some representative dimensions on FB15K237.

| | | 10d | | 20d | | 40d | | 80d | | 160d | | 320d | | 640d | |
|---|---|---|---|---|---|---|---|---|---|---|---|---|---|---|---|
| | *Method* | *Hit@3* | *Hit@1* | *Hit@3* | *Hit@1* | *Hit@3* | *Hit@1* | *Hit@3* | *Hit@1* | *Hit@3* | *Hit@1* | *Hit@3* | *Hit@1* | *Hit@3* | *Hit@1* |
| **TransE** | DT | 0.169 | 0.102 | 0.301 | 0.190 | 0.327 | 0.212 | 0.340 | 0.218 | 0.348 | 0.222 | 0.353 | 0.224 | **0.358** | **0.228** |
| | Ext | 0.123 | 0.065 | 0.211 | 0.122 | 0.264 | 0.156 | 0.296 | 0.180 | 0.320 | 0.197 | 0.331 | 0.208 | 0.358 | 0.228 |
| | Ext-L | 0.118 | 0.065 | 0.192 | 0.115 | 0.256 | 0.157 | 0.292 | 0.180 | 0.316 | 0.198 | 0.333 | 0.210 | 0.358 | 0.228 |
| | Ext-V | 0.150 | 0.081 | 0.222 | 0.126 | 0.265 | 0.156 | 0.301 | 0.185 | 0.325 | 0.205 | 0.341 | 0.217 | 0.358 | 0.228 |
| | BKD | 0.178 | 0.106 | 0.308 | 0.198 | 0.336 | 0.208 | 0.349 | 0.222 | 0.349 | 0.223 | 0.354 | 0.226 | - | - |
| | TA | 0.188 | 0.112 | 0.307 | 0.200 | 0.336 | 0.212 | 0.348 | 0.220 | 0.353 | 0.225 | 0.355 | 0.223 | - | - |
| | DualDE | 0.193 | 0.115 | 0.307 | **0.201** | 0.337 | 0.216 | **0.351** | 0.223 | **0.354** | 0.226 | 0.356 | 0.227 | - | - |
| | IterDE | 0.187 | 0.112 | 0.299 | 0.185 | 0.333 | 0.214 | 0.351 | 0.222 | 0.353 | 0.223 | 0.354 | 0.224 | - | - |
| | MED | **0.215** | **0.122** | **0.321** | 0.199 | **0.338** | **0.218** | 0.347 | **0.223** | 0.351 | **0.226** | **0.356** | **0.227** | 0.358 | 0.227 |
| | *Method* | *Hit@3* | *Hit@1* | *Hit@3* | *Hit@1* | *Hit@3* | *Hit@1* | *Hit@3* | *Hit@1* | *Hit@3* | *Hit@1* | *Hit@3* | *Hit@1* | *Hit@3* | *Hit@1* |
| **SimplE** | DT | 0.103 | 0.055 | 0.193 | 0.105 | 0.256 | 0.161 | 0.297 | 0.191 | 0.314 | 0.197 | 0.323 | 0.208 | **0.324** | **0.211** |
| | Ext | 0.039 | 0.019 | 0.071 | 0.047 | 0.091 | 0.057 | 0.171 | 0.109 | 0.251 | 0.159 | 0.294 | 0.187 | 0.324 | 0.211 |
| | Ext-L | 0.043 | 0.035 | 0.048 | 0.037 | 0.111 | 0.040 | 0.131 | 0.093 | 0.216 | 0.134 | 0.281 | 0.177 | 0.324 | 0.211 |
| | Ext-V | 0.047 | 0.036 | 0.074 | 0.043 | 0.097 | 0.077 | 0.145 | 0.109 | 0.248 | 0.156 | 0.289 | 0.189 | 0.324 | 0.211 |
| | BKD | 0.123 | 0.064 | 0.201 | 0.115 | 0.261 | 0.164 | 0.299 | 0.191 | 0.308 | 0.202 | 0.318 | **0.213** | - | - |
| | TA | 0.133 | 0.073 | 0.210 | 0.123 | 0.276 | 0.175 | 0.302 | 0.195 | 0.318 | 0.203 | 0.323 | 0.211 | - | - |
| | DualDE | 0.130 | 0.071 | 0.224 | 0.115 | 0.279 | 0.175 | 0.305 | 0.196 | 0.324 | **0.208** | **0.326** | 0.211 | - | - |
| | IterDE | 0.132 | 0.069 | 0.217 | 0.118 | 0.276 | 0.174 | 0.303 | 0.192 | **0.326** | 0.204 | 0.324 | 0.212 | - | - |
| | MED | **0.164** | **0.073** | **0.254** | **0.158** | **0.288** | **0.177** | **0.305** | **0.196** | 0.319 | 0.205 | 0.318 | 0.209 | 0.322 | 0.209 |
| | *Method* | *Hit@3* | *Hit@1* | *Hit@3* | *Hit@1* | *Hit@3* | *Hit@1* | *Hit@3* | *Hit@1* | *Hit@3* | *Hit@1* | *Hit@3* | *Hit@1* | *Hit@3* | *Hit@1* |
| **RotatE** | DT | 0.284 | 0.168 | 0.330 | 0.207 | 0.346 | 0.223 | 0.352 | 0.224 | 0.353 | 0.229 | 0.357 | 0.230 | **0.363** | **0.234** |
| | Ext | 0.152 | 0.080 | 0.225 | 0.129 | 0.278 | 0.170 | 0.304 | 0.190 | 0.322 | 0.203 | 0.335 | 0.217 | 0.363 | 0.234 |
| | Ext-L | 0.147 | 0.078 | 0.209 | 0.121 | 0.247 | 0.146 | 0.275 | 0.166 | 0.312 | 0.193 | 0.333 | 0.209 | 0.363 | 0.234 |
| | Ext-V | 0.174 | 0.097 | 0.218 | 0.126 | 0.264 | 0.159 | 0.293 | 0.182 | 0.319 | 0.201 | 0.336 | 0.213 | 0.363 | 0.234 |
| | BKD | 0.306 | 0.193 | 0.338 | 0.214 | 0.352 | 0.224 | 0.354 | 0.230 | 0.356 | 0.230 | 0.358 | 0.231 | - | - |
| | TA | 0.308 | 0.196 | 0.339 | 0.216 | 0.353 | 0.225 | 0.358 | 0.229 | 0.359 | 0.229 | 0.358 | 0.231 | - | - |
| | DualDE | 0.311 | 0.197 | 0.341 | 0.216 | 0.353 | **0.227** | **0.360** | 0.230 | 0.361 | 0.232 | 0.361 | 0.233 | - | - |
| | IterDE | 0.307 | 0.195 | 0.342 | 0.215 | 0.355 | 0.225 | 0.359 | 0.232 | **0.363** | 0.233 | 0.362 | **0.234** | - | - |
| | MED | **0.324** | **0.201** | **0.344** | 0.216 | **0.355** | 0.225 | 0.357 | 0.231 | 0.358 | **0.233** | 0.362 | 0.233 | 0.362 | 0.232 |
| | *Method* | *Hit@3* | *Hit@1* | *Hit@3* | *Hit@1* | *Hit@3* | *Hit@1* | *Hit@3* | *Hit@1* | *Hit@3* | *Hit@1* | *Hit@3* | *Hit@1* | *Hit@3* | *Hit@1* |
| **PairRE** | DT | 0.198 | 0.116 | 0.262 | 0.162 | 0.312 | 0.202 | 0.337 | 0.222 | 0.352 | 0.227 | 0.364 | 0.235 | **0.368** | **0.237** |
| | Ext | 0.158 | 0.107 | 0.187 | 0.118 | 0.236 | 0.149 | 0.283 | 0.182 | 0.325 | 0.207 | 0.354 | 0.230 | 0.368 | 0.237 |
| | Ext-L | 0.159 | 0.099 | 0.196 | 0.134 | 0.238 | 0.159 | 0.298 | 0.188 | 0.342 | 0.219 | 0.359 | 0.233 | 0.368 | 0.237 |
| | Ext-V | 0.181 | 0.116 | 0.192 | 0.125 | 0.250 | 0.154 | 0.307 | 0.193 | 0.343 | 0.221 | 0.362 | 0.237 | 0.368 | 0.237 |
| | BKD | 0.215 | 0.132 | 0.265 | 0.168 | 0.314 | 0.203 | 0.343 | 0.224 | 0.355 | 0.233 | 0.366 | 0.236 | - | - |
| | TA | 0.226 | 0.139 | 0.291 | 0.182 | 0.316 | 0.210 | 0.347 | 0.224 | 0.358 | 0.232 | 0.368 | 0.235 | - | - |
| | DualDE | 0.224 | 0.139 | 0.286 | 0.179 | 0.318 | 0.212 | 0.351 | 0.226 | **0.359** | **0.234** | **0.371** | **0.238** | - | - |
| | IterDE | 0.225 | 0.135 | 0.293 | 0.185 | 0.324 | 0.212 | **0.352** | 0.224 | 0.357 | 0.234 | 0.369 | 0.236 | - | - |
| | MED | **0.253** | **0.172** | **0.299** | **0.189** | **0.327** | **0.213** | 0.346 | **0.224** | 0.357 | 0.232 | 0.366 | 0.236 | 0.368 | 0.235 |

Table 13: Ablation study on WN18RR with TransE.

| dim | MED | | | | MED w/o MLM | | | | MED w/o EIM | | | | MED w/o DLW | | | |
|---|---|---|---|---|---|---|---|---|---|---|---|---|---|---|---|---|
| | *MRR* | *Hit@10* | *Hit@3* | *Hit@1* | *MRR* | *Hit@10* | *Hit@3* | *Hit@1* | *MRR* | *Hit@10* | *Hit@3* | *Hit@1* | *MRR* | *Hit@10* | *Hit@3* | *Hit@1* |
| 10 | .170 | **.388** | **.269** | .036 | .149 | .335 | .234 | .032 | .169 | .388 | .267 | **.037** | **.171** | .387 | .268 | .035 |
| 20 | **.219** | **.491** | **.369** | .042 | .197 | .437 | .323 | .032 | .217 | .488 | .366 | **.044** | .218 | .487 | .367 | .039 |
| 40 | **.232** | **.518** | .399 | **.048** | .224 | .496 | .379 | .029 | .232 | .517 | **.403** | .042 | .232 | .517 | .402 | .037 |
| 80 | .232 | .523 | .404 | **.042** | .228 | .521 | .399 | .033 | **.235** | **.529** | .408 | .037 | .234 | .523 | **.410** | .041 |
| 160 | **.236** | **.529** | **.407** | .037 | .234 | .525 | .406 | .032 | .234 | .527 | .405 | .032 | .235 | .527 | .405 | .032 |
| 320 | **.237** | **.536** | **.410** | .033 | .236 | .532 | .409 | **.035** | .233 | .530 | .398 | .031 | .234 | .533 | .405 | .029 |
| 640 | .237 | **.537** | **.412** | .031 | **.238** | .535 | .412 | **.042** | .232 | .528 | .402 | .029 | .233 | .530 | .396 | .025 |

existence of the mutual learning mechanism, the low-dimensional sub-model can still learn from the high-dimensional sub-model, so as to ensure the certain performance of the low-dimensional sub-model. In addition, we also find that as the dimension increases to a certain extent, the performance of the sub-model does not improve, and even begins to decline. We guess that this is because the mutual learning mechanism makes every pair of neighbor sub-models learn from each other, resulting in some low-quality or wrong knowledge gradually transferring from the low-dimensional sub-models to the high-dimensional sub-models, and when the evolutionary improvement mechanism is removed, the high-dimensional sub-models can no longer correct the wrong information from the low-dimensional sub-models. The higher the dimension of the sub-model, the more the accumulated error, so the performance of the high-dimensional sub-models is seriously damaged. On the whole, this mechanism mainly helps to improve the effect of high-dimensional sub-models.

### B.3 DYNAMIC LOSS WEIGHT (DLW)

To study the effect of the dynamic loss weight, we fix the ratio of all mutual learning losses to all evolutionary improvement losses as $1 : 1$, and equation 6 is rewritten as

$$L = \sum_{i=2}^{n} L_{ML}^{i-1,i} + \sum_{i=1}^{n} L_{EI}^{i}. \tag{9}$$

According to the result of "MED w/o DLW" in Table 13, the overall results of "MED w/o DLW" are in the middle of the results of "MED w/o MLM" and "MED w/o EIM": the performance of the low-dimensional sub-model is better than that of "MED w/o MLM", and the performance of the high-dimensional sub-model is better than that of "MED w/o EIM". On the whole, its results are more similar to "MED w/o EIM", that is, the performance of the low-dimensional sub-model does not change much, while the performance of the high-dimensional sub-model decreases more significantly. We believe that for the high-dimensional sub-model, the proportion of mutual learning loss is still too large, which makes it more negatively affected by the low-dimensional sub-model. This result indicates that the dynamic loss weight plays a role in adaptively balancing multiple losses and contributes to improving overall performance.

## C DETAILS OF APPLYING THE TRAINED KGE BY MED TO REAL APPLICATIONS

The SKG is used in many tasks related to users, and injecting user embeddings trained over SKG into downstream task models is a common and practical way.

User labeling is one of the common user management tasks that e-commerce platforms run on backend servers. We model user labeling as a multiclass classification task for user embeddings with a 2-layer MLP:

$$\mathcal{L} = -\frac{1}{|\mathcal{U}|} \sum_{i=1}^{|\mathcal{U}|} \sum_{j=1}^{|\mathcal{CLS}|} y_{ij} \log(\text{MLP}(u_i)), \tag{10}$$

where $u_i$ is the $i$-th user's embedding, the label $y_{ij} = 1$ if user $u_i$ belongs to class $cls_j$, otherwise $y_{ij} = 0$.

The product recommendation task is to properly recommend items to users that users will interact with a high probability and it often runs on terminal devices. Following PKGM Zhang et al. (2021), which recommends items to users using the neural collaborative filtering (NCF) He et al. (2017) framework with the help of pre-trained user embeddings as service vectors, we add trained user embeddings over SKG as service vectors to NCF. In NCF, the MLP layer is used to learn item-user interactions based on the latent feature of the user and item, that is, for a given user-item pair $user_i - item_j$, the interaction function is

$$\phi_1^{MLP}(p_i, q_j) = \text{MLP}([p_i; q_j]), \tag{11}$$

where $p_i$ and $q_j$ are latent feature vectors of user and item learned in NCF. We add the trained user embedding $u_i$ to NCF's MLP layer and rewrite Equation equation 11 as

$$\phi_1^{MLP}(p_i, q_j, u_i) = \text{MLP}([p_i; q_j; u_i]), \tag{12}$$

and the other parts of NCF stay the same as in PKGM Zhang et al. (2021).

We train entity and relation embeddings for SKG based on TransE Bordes et al. (2013) and input the trained entity (user) embedding into Equation equation 10 and Equation equation 12.

# D  DETAILS OF EXTENDING MED TO LANGUAGE MODEL BERT-BASE

## D.1  DATASET AND EVALUATION METRIC

For the experiments extending MED to BERT, we adopt the common GLUE Wang et al. (2019) benchmark for evaluation. To be specific, we use the development set of the GLUE benchmark which includes four tasks: Paraphrase Similarity Matching, Sentiment Classification, Natural Language Inference, and Linguistic Acceptability. For Paraphrase Similarity Matching, we use MRPC Dolan & Brockett (2005), QQP and STS-B Conneau & Kiela (2018) for evaluation. For Sentiment Classification, we use SST-2 Socher et al. (2013). For Natural Language Inference, we use MNLI Williams et al. (2018), QNLI Rajpurkar et al. (2016), and RTE for evaluation. In terms of evaluation metrics, we follow previous work Devlin et al. (2019); Sun et al. (2019a). For MRPC and QQP, we report F1 and accuracy. For STS-B, we consider Pearson and Spearman correlation as our metrics. The other tasks use accuracy as the metric. For MNLI, the results of MNLI-m and MNLI-mm are both reported separately.

## D.2  BASELINES

For comparison, we choose Knowledge Distillation (KD) models and Hardware-Aware Transformers Wang et al. (2020a) (HAT) customized for transformers as baselines. For the KD models, we compare MED with Basic KD (BKD) Hinton et al. (2015), Patient KD (PKD) Sun et al. (2019a), Relational Knowledge Distillation (RKD) Park et al. (2019), Deep Self-attention Distillation (MiniLM) Wang et al. (2020b), Meta Learning-based KD (MetaDistill) Zhou et al. (2022a) and Feature Structure Distillation (FSD) Jung et al. (2023). For the comparability of the results, we choose 4-layer BERT (BERT$_4$) or 6-layer BERT (BERT$_6$) as the student model architectures, which guarantees that the number of model parameters (*#P(M)*) or *speedup* is comparable. For HAT, we use the same model architecture as our MED for training and show the results of sub-models with three parameter scales.

## D.3  IMPLEMENTATION

To implement MED on BERT, for the word embedding layer, all sub-models share the front portion of embedding parameters in the same way as in KGE, and for the transformer layer, all sub-models share the front portion of weight parameters as in HAT Wang et al. (2020a). Specifically, assuming that the embedding dimension of the largest BERT model $B_n$ is $d_n$, and the embedding dimension of the sub-model $B_i$ is $d_i$, for any parameter matrix with the shape $x \times y$ in $B_n$, the front portion sub-matrix of it with the shape $\frac{d_i}{d_n}x \times \frac{d_i}{d_n}y$ is the parameter matrix of the corresponding position in $B_i$. Finally, it just need to replace the triple score $s_{(h,r,t)}$ in Equation equation 2, Equation equation 3, Equation equation 4, and Equation equation 5 with the logits output for the corresponding category of the classifier in the classification task.

We set $n = 4$ for BERT applying MED, and 4 sub-models have the following settings: [768, 512, 256, 128] for embedding dim and [768, 512, 256, 128] for hidden dim, [12, 12, 6, 6] for the head number in attention modules, 12 for encoder layer number.

