# OpenReview forum: "Croppable Knowledge Graph Embedding"
_ICLR.cc/2025/Conference — Submitted to ICLR 2025_

### Official Review · Reviewer_puCW · 2024-11-01

**Soundness:** 3
**Presentation:** 3
**Contribution:** 2
**Rating:** 5
**Confidence:** 4

**Summary:**

In conventional KGE approaches, each change in embedding dimensions necessitates retraining the entire model from the beginning. This paper aims to train a single KGE model that can be "cropped" into sub-models of various dimensions without additional training, thus reducing the overhead and enhancing the flexibility of KGE applications.

**Strengths:**

1.The idea of the croppable KGE is interesting and makes sense.
2.The manuscript is well-organized and easy to follow
3.Authors provide extensive experimental results, especially on real-world applications and LMs

**Weaknesses:**

1.The authors claim one of major contributions is that “the training efficiency of MED is far higher than that of independently training multiple KGE models of different sizes or obtaining them by knowledge distillation.” This comparison seems unfair because the multiple models in MED are parameter sharing. I expect to see some impressive techniques to reduce redundant gradient calculations, instead of a rough procedure of iterative model-pair training. This makes the technical contribution limited.

2.In experimental settings, configuring 64 sub-models represents a significant investment in resources. However, in practical applications, training such a large number of models with varying dimensions is often unnecessary. Typically, it is adequate to develop models for a few key dimension settings that cater to the requirements of servers, PCs, and mobile devices. While increasing the number of sub-models might enhance training performance, this approach can be excessively time-consuming and resource-intensive in real-world scenarios. Therefore, it would be more convincing to evaluate the effectiveness and efficiency of the multi-submodel MED by comparing it against the traditional method of training three specifically dimensioned models tailored to the aforementioned application settings.

3.In MED, there are many learnable weights. It would be useful to know about the robustness of the training and the time it takes for the training to converge. Providing a curve showing how the training loss/metric changes over time would be even more informative.

4.Compared to training 64 sub-models, I am curious about the performance when training different numbers of sub-models. (No need to provide comprehensive results in the rebuttal phase.)

**Questions:**

Please refer to the Weaknesses.

---

> ### Author Response · Authors · 2024-11-20
>
> Dear Reviewer:
>
> Thank you for your insightful feedback. We have thoughtfully addressed each of your comments and questions. We hope our response meets you in good spirits and enhances your overall impression of our paper.
>
> **W1: About technical contribution**
>
> **R:** Our main technical contribution lies in the design of the parameter-sharing pattern among multiple sub-models and the provision of a training scheme that focuses on how to simultaneously train multiple sub-models while ensuring all of their performance.
>
> Ensuring the availability of all models at the same time is very difficult. We solve this problem by proposing a training framework consisting of three modules: mutual learning mechanism, evolutionary improvement mechanism, and dynamic loss weight. As in Appendix B ABLATION STUDY, for the three modules we proposed, any lack of one leads to a decrease in the overall performance of the models.
>
> Finally, our method not only significantly improves the performance of low-dimensional models but also guarantees good performance of high-dimensional models. Even in the real application experiment, only three models for different devices are required. The best baseline requires 3.7 times more training time than our method (Table 5) and has worse performance than our method.
>
> Therefore, the increase in training efficiency brought about by this sharing of parameters is also one of our advantages and contributions.
>
> **W2: Experiments in practical applications**
>
> **R:** We set as many models as possible in the main experiment, aiming to verify whether our method can meet as many dimensional requirements as possible.
>
> In actual application scenarios, refer to Section 5.4 MED IN REAL APPLICATIONS. For applications with three different dimensions (10d for mobile phone, 100d for PC, and 500d for server), only three sub-models with the required dimensions are set. Our method achieves the best performance: The ndcg@5 of our models on the product recommendation task exceeded the optimal baseline by 4.5% for 10d and 2.1% for 100d. The f1 of our models on the user labeling task exceeded the optimal baseline by 0.6%.
>
> In addition, we have added the comparison of training time of different methods in "Table 5: Results on SKG" in our new version (highlighted in red in the pdf). The conclusion is that the training efficiency of our method is 3.7 times faster than the optimal baseline.
>
> **W3: Added a curve showing training metric**
>
> **R:** Thanks for your suggestion, in Section 5.6.1 TRAINING EFFICIENCY, we have added a curve showing the variation of sub-models' MRR during training (Figure 4) in our new version (highlighted in red in the pdf).
>
> Figure 4 shows that high-dimensional models (160d, 640d) achieve better performance at the first 100 epochs, but they converge more slowly, requiring about 1500 epochs. Low-dimensional models (10d, 40d) converge faster than high-dimensional models, requiring less than 1000 epochs.
>
> **W4: Performance of different numbers of sub-models**
>
> **R:** We have added Section 5.6.2 EFFECT OF THE NUMBER OF SUB-MODELS in the new version (highlighted in red in the pdf). On RotatE and WN18RR, we set different numbers of sub-models: n=64, n=16, and n=4. All of these include sub-models of dimensions 10d, 40d, 160d, and 640d.
>
> The results in Table 8 show that as n decreases, the performance of high-dimensional sub-models (160d, 640d) improves, while the performance of low-dimensional (10d, 40d) models decreases. However, our low-dimensional models still exceed the best baselines in Table 3: MRR=0.299 for 10d Ext and MRR=0.462 for 40d DualDE.
>
> The training efficiency is almost linearly related to the number of models. The training time required for n=64, n=16, and n=4 is 3.3h, 6.2h, and 12.7h, respectively.
>
> The results prove that in practical application scenarios, the performance and efficiency of our method have advantages regardless of the number of models required.
>
>
> - Table 8: Results of different n.
>
> |     |              | **10d** |       | **40d** |       | **160d** |       | **640d** |       |
> |:---:|:------------:|:-------:|:-----:|:-------:|:-----:|:--------:|:-----:|:--------:|:-----:|
> | _n_ | _train time_ |  _MRR_  | _H10_ |  _MRR_  | _H10_ |   _MRR_  | _H10_ |   _MRR_  | _H10_ |
> |  64 |     12.7h    |  0.324  | 0.469 |  0.466  | 0.561 |   0.471  | 0.574 |   0.476  | 0.574 |
> |  16 |     6.2h     |  0.322  | 0.467 |  0.465  | 0.561 |   0.473  | 0.575 |   0.477  | 0.576 |
> |  4  |     3.3h     |  0.319  | 0.463 |  0.463  | 0.561 |   0.475  | 0.577 |   0.480  | 0.578 |

---

### Official Review · Reviewer_NV9A · 2024-11-04

**Soundness:** 3
**Presentation:** 3
**Contribution:** 2
**Rating:** 6
**Confidence:** 4

**Summary:**

This paper introduces a croppable knowledge graph embedding training framework MED. This MED is consisting of three modules. The first mutual learning mechanism is mutual learning mechanism that is used to make the two submodels learn from each other. The second evolutionary improvement focuses on enabling the high-dimensional submodel can get the knowledge that low-dimensional model cannot predict correctly. The final mechanism is the dynamic loss weight for balancing the multiple losses of submodels.

**Strengths:**

Strengths:

1.	This paper presents an interesting problem that is how to train a croppable KGE so that more different dimensions can be cropped from the embeddings. The whole idea is clear and easy to follow in this paper.

2.	A framework MED is proposed for serving the purpose of croppable embeddings. This framework is consisting of multiple sub-models. The low dimensional models are similar to our original KGE. Authors want to improve the performance as much as possible. The high dimensional models are more different because they need to master the knowledge that low-dimensional sub models cannot learn well. Here the “master” is not easy to understand. It is designed for high-dimensional model can make a better prediction based on the prediction of low-dimensional model based on the evolutionary improvement mechanism.

**Weaknesses:**

Weaknesses:

1.	The ablation studies are needed for this paper. The MED includes three modules: mutual learning mechanism, evolutionary improvement mechanism and a dynamic loss weight. It is very important to evaluate the effectiveness of each module and discuss if there is any alternative solution here. For instance, can we just duplicate a model that dimension d is small num n times and use the evolutionary improvement mechanism to tune them for satisfying the target that high-dimensional models need to master the knowledge that low-dimensional model cannot predict.

2.	It is not clear about how to choose the dimension size of each sub-model and define the number of sub-models.

3.	For the mutual learning mechanism, the purpose is to make two models learn from each other, while the evolutionary improvement mechanism is to make high dimensional model learns more knowledge that low-dimensional model cannot predict. Will the purposes of these two losses opposite？

**Questions:**

Please check the weaknesses and answer the questions.

---

> ### Author Response · Authors · 2024-11-20
>
> Dear Reviewer:
>
> Thank you for your insightful feedback. We have thoughtfully addressed each of your comments and questions. We hope our response meets you in good spirits and enhances your overall impression of our paper.
>
> **W1: Ablation studies are needed**
>
> **R:** Due to the page limitation of the main text, we put the ablation study in the appendix. In Appendix B ABLATION STUDY, we analyzed in detail the role of each module and their effect on sub-models of different sizes.
>
> The conclusion is that the mutual learning mechanism mainly guarantees the performance of low-dimensional models. The evolutionary improvement mechanism is more conducive to improving the performance of high-dimensional models. The dynamic loss weight module can indirectly improve the weight of mutual learning losses by reducing the evolutionary loss for low-dimensional models. It can also indirectly reduce the weight of mutual learning loss by increasing the evolutionary loss for high-dimensional models. Any lack of one of the three modules makes these models' overall performance decrease.
>
> In addition, we tried a similar approach to the one you mentioned in our early model design: first train the lowest dimensional model, then replicate and fix those dimensions into a high-dimensional model, and then use the evolutionary improvement mechanism to tune high-dimensional models. However, the performance of high-dimensional models by this method is very poor.
>
> **W2: How to choose the dimension size of each sub-model and number of model**
>
> **R:** The number of sub-models and the model size should be selected according to the actual application. We set as many models as possible in the main experiment, aiming to verify whether our method can meet as many dimensional requirements as possible.
>
> In actual application scenarios, referring to Section 5.4 MED IN REAL APPLICATIONS, for applications with three different dimensions (10d for mobile phone, 100d for PC, and 500d for server), only three sub-models with the required dimensions are set.
>
> The experimental results show that three sub-models by our method all achieve the best performance in user labeling and product recommendation tasks: The ndcg@5 of our models on the product recommendation task exceeded the optimal baseline by 4.5% for 10d and 2.1% for 100d. The f1 of our models on the user labeling task exceeded the optimal baseline by 0.6%. In addition, we have added the comparison of training time of different methods in "Table 5: Results on SKG" in our new version (highlighted in red in the pdf). The conclusion is that the training efficiency of our method is 3.7 times faster than the optimal baseline.
>
> **W3: The mutual learning mechanism and evolutionary improvement mechanism opposite**
>
> **R:**  Intuitively, yes. Mutual learning helps low-dimensional models to learn better from high-dimensional models. The evolutionary improvement mechanism helps high-dimensional models to improve based on low-dimensional models.
>
> We suppose that the opposite you mentioned means that the distillation loss of mutual learning (Eq.1) affects both high-dimensional and low-dimensional models. If the weight of Eq.1 is too large, it will weaken the improvement brought by evolutionary improvement loss (Eq.5) for high-dimensional models.
>
> This conflict does exist. We solve this by a learnable scaler coefficient in the third module "dynamic loss weight." It adaptively assigns the weights of Eq.1 and Eq.5 to different sub-models. For high-dimensional models, the weights of Eq.5 are greater to reduce the negative effects from Eq.1.
>
> The results in Appendix B ABLATION STUDY also proved that the dynamic loss weight module plays a great role in obtaining good overall performance of all models by assigning different weights of mutual learning loss and evolutionary improvement loss to different submodels.

---

> > ### Comment · Reviewer_NV9A · 2024-12-02
> >
> > Thanks a lot for authors' response and explaining details to address my concerns. The ablation studies explain the impact of each module, which proves the contributions of all modules. However, the conflict of two losses is still a concern for me. Optimizing two opposite tasks will be a potential problem. It will be great to discuss this more in the new version. After considering the all factors, I'd like to keep my score unchanged.

---

### Official Review · Reviewer_1ZDq · 2024-11-04

**Soundness:** 2
**Presentation:** 2
**Contribution:** 1
**Rating:** 3
**Confidence:** 5

**Summary:**

This paper addresses the challenge of efficiently adapting Knowledge Graph Embedding (KGE) models to various dimensional requirements without retraining from scratch. It introduces a novel KGE training framework called MED, which allows for the extraction of sub-models with specific dimensions from a single trained model, utilizing a mutual learning mechanism and adaptive loss weights. The results demonstrate that MED enhances performance across multiple KGE models and application scenarios, providing greater efficiency and flexibility compared to traditional methods.

**Strengths:**

1. The proposed method aims to train once to get a croppable KGE model applicable to multiple scenarios with different dimensional requirements, which is an interesting topic.
2. The authors improve the low-dimensional sub-model's performance and make the high-dimensional sub-models retain the capacity that low-dimensional sub-models have, which seems reasonable.

**Weaknesses:**

1. The paper is not organized clearly, which is not friendly for understanding. For example, there is a lack of preliminary details on how the previous knowledge distillation methods do.

2. The novelty of this paper seems limited since knowledge distillation has already been used in the previous work [1].
[1] Lifelong embedding learning and transfer for growing knowledge graphs

3. The paper lacks the analysis of time complexity as well as space complexity, which is necessary to study the efficiency of the model.
4. The authors do not compare the model with other SOTA KGE methods, e.g.,[1][2][3]. The performance of, MRR of these models in FB15K-237 is 0.36 while that of the proposed paper is 0.323. In this way, the performance of the proposed paper is not significant and the authors may better give a reasonable explanation.
[1] Compounding Geometric Operations for Knowledge Graph Completion
[2] Geometry interaction knowledge graph embeddings
[3] KRACL: Contrastive Learning with Graph Context Modeling for Sparse Knowledge Graph Completion

**Questions:**

Please refer to weaknesses.

---

> ### Author Response · Authors · 2024-11-20
>
> Dear Reviewer:
>
> Thank you for your insightful feedback. We have thoughtfully addressed each of your comments and questions. We hope our response meets you in good spirits and enhances your overall impression of our paper.
>
> **W1: Lack of preliminary details**
>
> **R:** Instead of a preliminary section, we include the introduction of how previous knowledge distillation methods do in the Related Work Section with proper references, specifically referring to 2.2 KNOWLEDGE DISTILLATION. Further more, to make it clear how the knowledge distillation baseline methods we selected in the experiments do, we also introduce them in detail in Section 5.1.4 BASELINES including the reference of these works. We believe this should be enough for readers to understand how previous knowledge distillation methods do.
>
> **W2: Novelty is limited since knowledge distillation is used in the previous work**
>
> **R:** We think our work is quite different from the previous work you mentioned. We do not involve lifelong embedding, and the KG remains unchanged in our setting. Our novelty lies in proposing a KGE training framework called MED. This framework trains once to get a croppable KGE model applicable to multiple scenarios with multi-dimension requirements. Sub-models of the required dimensions can be cropped out of it and used directly without any additional training.
>
> We design a parameter-sharing pattern among multiple sub-models and provide a training scheme that focuses on how to simultaneously train multiple sub-models while ensuring their usability. Knowledge distillation, as an effective technique, can be applied in various fields of KG, including lifelong learning [1], model compression [2][3], and transfer learning [4].
>
> In our methods, knowledge distillation is one of the techniques used to implement our framework, embodied in the mutual learning mechanism. In addition, we have two other important modules: the evolutionary improvement mechanism and dynamic loss weight. The three modules work together to ensure the availability and validity of all sub-models in the entire framework.
>
> [1] Lifelong embedding learning and transfer for growing knowledge graphs
>
> [2] Dualde: Dually distilling knowledge graph embedding for faster and cheaper reasoning.
>
> [3] Iterde: An iterative knowledge distillation framework for knowledge graph embeddings.
>
> [4] Mulde: Multi-teacher knowledge distillation for low-dimensional knowledge graph embeddings.
>
> **W3: Lacks the analysis of time complexity and space complexity**
>
> **R:** We analyzed the time complexity of training in Section 5.6.1 TRAINING EFFICIENCY. The conclusion is that our approach achieved a time efficiency improvement of about 10 times, as shown in Table 7.
>
> We also analyzed the space complexity of parameters in Section 5.3 PARAMETER EFFICIENCY OF MED. The conclusion is that our method achieved a space efficiency improvement of 54.1% and 25.9% on datasets WN18RR and FB15K237, respectively, as shown in Table 4.
>
> In addition, we have added the comparison of training time of different methods in "Table 5: Results on SKG" in Section 5.4 MED IN REAL APPLICATIONS in our new version (highlighted in red in the pdf). The comparison on real application tasks shows that the training efficiency of our method is 3.7 times faster than the optimal baseline and achieves the best performance.
>
> We also added Section 5.6.2 EFFECT OF THE NUMBER OF SUB-MODELS in the new version. We set different numbers of sub-models: n=64, n=16, and n=4. The results in Table 8 show that as n decreases, the training efficiency almost linearly increases. The training time required for n=64, n=16, and n=4 is 3.3h, 6.2h, and 12.7h, respectively.
>
> In conclusion, our method shows high time and space efficiency.
>
> **W4: Do not compare with other SOTA KGE methods**
>
> **R:** We propose a general approach applicable to various Knowledge Graph Embedding (KGE) models. Our method is applicable to any KGE method with a scoring function, including the one you mentioned in [2].
>
> As a research paper, we cannot conduct experiments on all KGE methods. To demonstrate the universality of our method, we selected representative models from different KGE types: distance-based TransE, rotation-based RotatE, semantic matching-based SimplE, and subrelation encoding-based PairRE.
>
> On these KGEs, our approach outperforms baselines in almost all settings. For example, on WN18RR with 10d, our method improves significantly over the best MRR of baselines: from 0.148 to 0.170 on TransE, 0.089 to 0.111 on SimplE, 0.299 to 0.324 on RotatE, and 0.245 to 0.315 on PairRE. For high-dimensional models, our method enables them to achieve results competitive with directly trained ones on all these KGEs, as shown in Figure 3 and Figure 7.
>
> Therefore, the results of four fully different types of KGE models can demonstrate the universality and validity of our method.

---

### Official Review · Reviewer_T2ZS · 2024-11-04

**Soundness:** 3
**Presentation:** 2
**Contribution:** 3
**Rating:** 6
**Confidence:** 4

**Summary:**

Knowledge Graph Embeddings (KGEs) project entities and relationships into a continuous vector space and are widely applied in tasks like link prediction. Typically, increasing the embedding dimension enhances performance, yet device capabilities and storage limitations often dictate the feasible dimensionality. This paper tackles this issue by introducing a novel training framework, named MED, designed to train adaptable KGEs that work across various dimensions suitable for different scenarios.
The MED framework includes several modules: a mutual learning mechanism, an evolutionary improvement mechanism, and dynamic weight loss. In the mutual learning mechanism, the smaller-dimension model acts as a teacher to the higher-dimension model (the student), helping the student model retain knowledge from the higher-dimensional embeddings. To further refine the higher-dimension model by addressing what the teacher model has missed, the evolutionary improvement mechanism leverages hard labels and a weighted loss approach. Dynamic loss weighting is achieved through a weighted combination of these losses.
The proposed MED method has been evaluated on multiple datasets and compared with various embedding techniques, including distillation-based approaches. Results demonstrate the effectiveness of MED in improving performance across different embedding dimensions.

**Strengths:**

– The core idea and motivation of the paper are sound, and such approaches are essential to support real-world applications.

– The experiments have been done across a wide range of datasets from smaller ones to large one (SKG). In addition, the authors show their approach is general and can be extended to other machine learning models such as BERT. Thus the model may have a high impact beyond KGE models.

– In very low dimension, e.g., 10d the method shows superior performance comparing to other models.

– the approach seems to be more efficient than previous approaches, e.g., knowledge distillation.

**Weaknesses:**

– In high dimension, the results are not better than other models in most cases.

– The technical contribution of the paper is not very significant. The main loss is combination of two existing losses. Moreover, the equation 4 is the same as equation 5 in the RotatE paper (the only difference is that equation 4 is used for two model, please add citation).

**Questions:**

Which patterns are learnt in high dimension that cannot be learned in low dimension?

---

> ### Author Response · Authors · 2024-11-20
>
> Dear Reviewer:
>
> Thank you for your insightful feedback. We have thoughtfully addressed each of your comments and questions. We hope our response meets you in good spirits and enhances your overall impression of our paper.
>
> **W1: Results of High dimension are not better than other models**
>
> **R:** As we analyzed in our Appendix B ABLATION STUDY, the reason why it is difficult to improve the performance of high-dimensional models is that the mutual learning mechanism may have negative effects on high-dimensional models.  This mechanism brings low-quality knowledge from low-dimensional sub-models.  However, the evolutionary improvement mechanism weakens this negative effect.  As a result, high-dimensional models still achieve competitive performance compared to directly trained KGEs, as shown in Figure 3.
>
> Furthermore, we believe that significantly enhancing the performance of low-dimensional models is more meaningful than marginally improving high-dimensional models. This is because current low-dimensional models often struggle with poor performance, making them impractical for real-world deployment and limiting their application on many edge devices.
>
> Our approach significantly improves the performance of low-dimensional models. For example, PairRE's MRR exceeds the current best baseline by up to 30% at 10 dimensions. In a real application experiment, as shown in Table 5, the ndcg@5 of our 10d model on the product recommendation task exceeded the optimal baseline by 4.5%.
>
> In contrast, high-dimensional models trained directly are already quite effective. The current baseline methods offer only limited improvements over direct training. For instance, at 160 dimensions, the best baseline on WN18RR only shows an average MRR increase of about 1% across four KGE models compared to direct training with TransE.
>
> We aim to provide a method for training models of different sizes simultaneously, with the challenge being to ensure the overall performance of all models.  Our method enables high-dimensional models to achieve results competitive with directly trained models.  This is sufficient to demonstrate the effectiveness of our approach.
>
> **W2: The main loss is combination of two existing losses**
>
> **R:** Loss design of Equation 4 has been widely used in various KGE methods and has been proven effective. We have added references to this in our new version (highlighted in red in the pdf).
>
> The specific individual loss design is not our main technical contribution.  Our main technical contribution lies in the design of the parameter-sharing pattern among multiple sub-models and the provision of a training scheme that focuses on how to simultaneously train multiple sub-models while ensuring their performance.
>
> Ensuring the availability of all models at the same time is very difficult.  As we saw in Appendix B ABLATION STUDY, for the three modules we proposed: mutual learning mechanism, evolutionary improvement mechanism, and dynamic loss weight, any lack of one can prevent these models from achieving better overall performance.
>
> **Q1: Patterns learnt in high dimension that cannot be learned in low dimension**
>
> **R:** When using the same KGE method, the patterns could be learned by different-size models are the same due to the same score function.   For example, based on RotatE, no matter how many dimensions of the final KGE model, symmetry, antisymmetry, inversion, and composition patterns are theoretically supported.  However, higher-dimensional models learn these patterns better.
>
> As can be seen from the entity nodes clustering (Figure 6 in Section 5.6.4 VISUAL ANALYSIS OF EMBEDDING), with the increase of dimension, the entity nodes of one type are more closely distributed and gradually differentiated into more small clusters, indicating that the high-dimensional model has learned higher-quality representation and mastered more fine-grained type information.

---

### Meta-Review · Area_Chair_5B5H · 2024-12-18

**Metareview:**

This manuscript proposes a Knowledge Graph Embedding (KGE) training framework named MED to address the challenge of efficiently adapting KGE models to various dimensional requirements without retraining from scratch. MED consists of three main modules: the Mutual learning mechanism, the Evolutionary improvement mechanism, and the Dynamic loss weight. The mutual learning mechanism improves the low-dimensional sub-models performance and makes the high-dimensional sub-models retain the capacity that low-dimensional sub-models have. The evolutionary improvement mechanism promotes the high-dimensional sub-models to master the knowledge that the low-dimensional sub-models can not learn, and the dynamic loss weight balances the multiple losses adaptively.

While the reviewers agreed on the paper's motivation, a croppable KGE model applicable to multiple scenarios with different dimensional requirements, they raised some critical concerns, mainly about the proposed method's limited novelty, unclear technical contributions and conflict of opposite task losses. The major comments from the reviewers are:
- The novelty of this paper seems limited since knowledge distillation has already been used in previous works. (Reviewer 1ZDq)
- The main loss is a combination of two existing losses. (Reviewer T2ZS)
- For the mutual learning mechanism, the purpose is to make two models learn from each other, while the evolutionary improvement mechanism is to make the high-dimensional model learn more knowledge that the low-dimensional model cannot predict. The conflict of these two losses is still a concern for me. Optimizing two opposite tasks will be a potential problem. (Reviewer NV9A)

**Additional Comments On Reviewer Discussion:**

While the reviewers agreed that this paper's topic is interesting, the authors should significantly revise the paper to resolve some concerns. Please find the details in the metareview. During the discussion period, Reviewer NV9A expressed the remaining concerns regarding the conflict of two opposite losses in the proposed method, but the authors did not engage in the discussion.

---

### Decision · Program_Chairs · 2025-01-22

Reject